# Possible climatic implications of high altitude emissions of black carbon

Gaurav Govardhan[1], Sreedharan Krishnakumari Satheesh[1,2], Ravi Nanjundiah[1,2], Krishnaswamy Krishna Moorthy[1], and Surendran Suresh Babu[3]

[1]Center for Atmospheric and Oceanic Sciences, Indian Institute of Science, Bengaluru, India
[2]Divecha Center for Climate Change, Indian Institute of Science, Bengaluru, India
[3]Space Physics Laboratory, Vikram Sarabhai Space Centre, Thiruvananthapuram, India

*Correspondence to:* Gaurav Govardhan (govardhan.gaurav@gmail.com)

**Abstract.** On account of its strong absorption of solar and terrestrial radiations, Black Carbon (BC) aerosol is known to impact large scale systems such as the Asian monsoon, Himalayan glaciers etc., besides affecting the thermal structure of lower atmosphere. While most studies focus on the near-surface abundance and impacts of BC, our study, using online regional chemistry transport model (WRF-Chem) simulations, examines the implications of sharp and confined layers of high BC concentration (called 'elevated BC layers') at altitudes more than 4 km over the Indian region, as revealed in the recent in-situ measurements using high-altitude balloons, carried out on $17^{th}$ March 2010, $8^{th}$ January 2011 and $25^{th}$ April 2011. Our study demonstrates that, the high-flying aircrafts (with emissions from the regionally fine-tuned MACCity inventory) are the most likely cause for these elevated BC layers. Furthermore, we show that such aircraft-emitted BC can get transported to even upper tropospheric/ lower stratospheric heights ($\sim$17 km) aided by the strong monsoonal convection occurring over the region, which are known to overshoot the tropical tropopause leading to injection of tropospheric air mass (along with its constituent aerosols) into the stratosphere. We show observational evidence for such an intrusion of tropospheric BC into the stratosphere over Indian region, using extinction coefficient and particle depolarization ratio data from CALIOP LIDAR on-board the CALIPSO satellite. We hypothesize that such intrusions of BC to lower stratosphere and its consequent longer residence time in the stratosphere would have significant implications for stratospheric ozone, considering the already reported ozone depleting potential of BC.

## 1 Introduction

The important role played by atmospheric aerosol particles in global and regional climate forcing is getting increasingly accepted. The global mean net radiative forcing (including the rapid adjustments) by aerosol is estimated to be -0.9 [-1.9 to -0.1] W m$^{-2}$ (Boucher et al., 2013), after accounting for the negative forcing by most of the aerosol species (sulphate, nitrate, sea-salts etc.) and the strong positive forcing by Black Carbon (BC), which absorbs solar radiation over a wide spectral band, and to a lesser extent by dust and organics. BC, a fine particulate, is formed as a result of incomplete combustion of fossil fuels, biofuels and biomass (Kuhlbusch et al., 1996; Ramanathan and Carmichael, 2008; Bond et al., 2013). Being a strong absorber of solar and terrestrial radiation, BC heats up the atmosphere (Petzold and Schönlinner, 2004; Ramanathan et al., 2007). The

global mean radiative forcing by BC is estimated as +0.4 (+0.05 to +0.8) W m$^{-2}$, which is around $(1/4)^{th}$ of the global mean CO$_2$ radiative forcing (Boucher et al., 2013). Several earlier studies have shown that atmospheric heating by a layer of aerosol with high BC abundance, is capable of perturbing large scale atmospheric phenomenon such as the Asian monsoon system and can also affect the hydrological cycle (Chakraborty et al., 2004; Ramanathan et al., 2005; Lau et al., 2006). Also, on ac-

count of its absorbing nature, BC, if deposited on snow, would exert 'snow-albedo forcing' resulting from the darkening of snow and could accelerate the melting of glaciers (Lau et al., 2010; Qian et al., 2011; Yasunari et al., 2010; Nair et al., 2013; Gautam et al., 2013). Furthermore, when BC particles are located above highly reflecting surfaces (such as snow/clouds), their absorption efficiency gets amplified (Haywood and Shine, 1995; Satheesh and Moorthy, 2005; Chand et al., 2009; Nair et al., 2013). This heating can give rise to local elevated dry convection, which can further lift the BC particles to higher levels of the

atmosphere (de Laat et al., 2012). The atmospheric heating by BC would be more severe at higher altitudes due to reduced air density at these altitudes, and this could amplify the heating effect. The high-altitude heating due to BC can also lead to reduced cloud cover at these elevated altitudes (Ackerman et al., 2000). Additionally, the atmospheric heating at higher altitudes can give rise to local stable scenario below, which can affect convection and consequently impact precipitation (Fan et al., 2008). Moreover, such lifted BC and the other pollutants, upon interaction with the strong monsoonal convection, can get lifted to

even higher heights (Andreae et al., 2001; Fromm et al., 2005; Randel et al., 2010; Vernier et al., 2011), and hold a potential to get lifted even beyond tropopause and intrude into the stratosphere under favorable upper tropospheric conditions. In this regard, the Tropical Tropopause Layer assumes importance. TTL is the region between the top of the major convective towers and the cold point. While forming an interface between the two dynamically different regimes, TTL acts as an entry-point to the tropical stratosphere (Fueglistaler et al., 2009). Sunilkumar et al. (2013) using radiosonde measurements, have reported

thinning of the Tropical Tropopause Layer (TTL) and reduction in the Temperature Lapse Rate (TLR) within TTL, which favor such cross-tropopause transport, over the Indian region during the monsoon season. A recent study by Das et al. (2016) reports a cross tropopause transport of air-mass over Indian region during the cyclone 'Nilam'. Once the particles enter the stratosphere, they reside for a longer period of time due to the inherent stability of stratosphere and the absence of strong removal mechanisms. Moreover, they can take part in heterogeneous chemical reactions taking place in stratosphere.

The aforementioned effects together highlight the significance of knowing the vertical distribution of BC especially over regions of strong convection (such as the Indian subcontinent especially during the pre-monsoon and the monsoon months). With this backdrop, a few studies have examined the vertical distribution of aerosols (extinction coefficient values and/or BC mass concentration values) over the Indian region using multiple observational platforms such as a ground based LIDAR (Satheesh et al., 2006), airborne and space based LIDAR (Satheesh et al., 2008; Prijith et al., 2016), instrumented aircrafts

(Moorthy et al., 2004; Babu et al., 2008, 2010; Nair et al., 2016), tethered and high-altitude balloons (Babu et al., 2011). Carrying out such observations for the first time over India, Moorthy et al. (2004) have noticed an exponential decrease of BC concentration within the boundary layer and a steady value above it, over an urban station (Hyderabad) in south-central India during the month of February, 2004. Examining the vertical profile of aerosols over Bangalore (an urban station in southern part of India) using a ground based micro pulse LIDAR, Satheesh et al. (2006) have reported morning-time (night-time) vertical

lifting (confinement) of aerosols during summer days. Carrying out airborne measurements of BC and absorption coefficient

over several stations across India, Babu et al. (2008, 2010); Nair et al. (2016) have reported a near-steady vertical distribution up to an altitude of 3 km above ground, with isolated peaks of higher BC concentration/ absorption coefficient during pre-monsoon period, above most of the landmass. Synthesizing multi-platform measurements Satheesh et al. (2008) have revealed the existence of elevated aerosol layers in the middle troposphere (4 to 5 km). They have shown that these elevated aerosol

layers caused atmospheric heating, which exhibited a meridional gradient that varied in vertical extent and amplitude from the northern Indian Ocean to Central India, during the pre-monsoon season. Using hydrogen inflated zero pressure balloon measurements for the first time in India, Babu et al. (2011) have revealed strikingly sharp and confined layers of BC at altitude of 4.5 km and 8 km over Hyderabad during the pre-monsoon season of 2010. They also have reported a large decrease in the environmental lapse rate associated with these peaks, using concurrent measurements aboard the same balloon. Analyzing

multi-year satellite (CALIPSO) data, Prijith et al. (2016), not only confirmed the existence of high altitude aerosol layers and their meridional gradient in the vertical extent during pre-monsoon periods over Indian region, but also have shown that mineral dust aerosol forms the dominant component of this layer. Kompalli et al. (2016) have reported spring-time enhancement in columnar Aerosol Optical Depth (AOD) and BC over Hanle - a western trans-Himalayan high-altitude observational location (4.5 km amsl). They attributed this enhancement to summer-time lifting of aerosols over the region west to south-west of Hanle

and subsequent long-range transport. All the above studies clearly bring out the existence of high altitude aerosol layers (with significant amount of BC) over the south Asian region during the pre-monsoon periods. Such strong absorbing aerosol layers can perturb the south Asian regional monsoonal system and can also affect the glacial coverage of the south Asian region. Thus, it becomes imperative to understand the dynamics of this regional high altitude aerosol layers and investigate the exact causes behind their occurrence.

In this study, we revisit the three high-altitude balloon measurements of BC made during the years 2010, and 2011, (in different seasons) over Hyderabad (the only location in India from where such balloon ascents could be made) and investigate the potential causes behind the existence of such confined BC layers, using a regional chemistry transport model. The balloon experiments, the model and simulation details are described section 2 and in section 3, we discuss the results. The possible climatic implications of such high-altitude BC layers are discussed in section 4. The conclusions are presented in section 5.

**2  Methods**

**2.1  High altitude measurements of BC**

The vertical profiles of BC used in this study have been obtained using an Aethalometer (Model AE-42, of Magee Scientific, USA), aboard high-altitude zero pressure balloons, launched from the National Balloon Facility at Hyderabad, ($17.48^0$N, $78.47^0$E, 557 m amsl), during three ascents made on 17 March 2010, $8^{th}$ January 2011 and $25^{th}$ April 2011. The first and

third flights corresponded to pre-monsoon conditions, while the second flight occurred in winter. The details of experimental set-up, calibration, data collection and analysis are available in Babu et al. (2011) and as such, only a brief account is given here. The 109755 $m^3$ zero-pressure balloon was made of 25 mm thick, linear low density polyethylene film, and was capable of carrying ∼350 kg of payload up to a ceiling altitude of ∼35 km. The mass concentration of 'Equivalent Black Carbon'

(EBC) (Petzold et al., 2013) was measured using an Aethalometer, which estimates the mass concentration of EBC by measuring the change in the transmittance of a quartz filter to 880 nm light upon the deposition of aerosol. The value of effective Mass Absorption Cross-section (MAC) used in these measurements is 16 $m^2$ $g^{-1}$ (Hansen, 1996, 2005). The effective MAC includes the amplification of absorption due to multiple scattering on the filter fiber matrix and the decrease due to shadowing.

The Aethalometer was configured for volumetric flow with an external pump, providing a flow rate of 14 liters per minute (LPM) at ground, and operated at a time base of 5 s. The data from the Aethalometer was telemetered down along with the GPS co-ordinates. A few studies have reported uncertainties in the Aethalometer estimated EBC (for e.g., Weingartner et al. (2003); Arnott et al. (2005); Sheridan et al. (2005); Corrigan et al. (2006)). Following the suggestions from the reports, we have used an amplification factor of 1.9 and an 'R' factor (shadowing effect) of 0.88 in this study. Nevertheless, when the aerosols

are aged or mixed with others, as they would be away from the direct emissions, the shadowing effect would be negligible (Weingartner et al., 2003). In addition to BC, dust also absorbs the short wave radiation, but Mass Absorption Cross-section (MAC) for dust is two to three orders of magnitude lesser than the for BC (Hansen, 2005). Hence if the mass of dust is substantially higher than EBC, only then same optical absorption will be produced. Thus under normal conditions, the effects of dust on measured EBC mass concentrations would be negligible.

Besides the Aethalometer, each ascent also carried other payloads such as GPS receiver, meteorological sensors for temperature and RH, telemetry and telecommand systems. The entire payload was tested using a thermo-vacuum facility to ensure consistent operations up to ∼9 km altitude (250 hPa pressure level and ∼ -40$^0$C). Beyond this altitude, as the ambient pressure drops too low to provide sufficient flow to the Aethalometer, the telecommand system was used to switch off the instrument for the higher altitudes, and switch it on again at this altitude during the descent phase. A gondola, carrying the payloads, had

been attached to the balloon using a parachute, which was deployed in the descent phase and enabled safe recovery of the payloads for reuse. Ballast cans attached to the gondola ensured a slow and steady ascent of ∼2.6 m $s^{-1}$, while the descent speed was controlled by the parachute, as the ballast cans were detached after the balloon rose to the free tropospheric altitude. The scientific data along with the housekeeping data were collected during the ascent and descent phases and were continuously telemetered to the ground, besides storing on-board. The total flight duration was about 3 to 4 hours and the spatial

ground spread of the flight path has been within 50 km radius of the launch site. The data have been analysed (following details in Babu et al. (2011)) and the profiles obtained were smoothed using a running mean filter. The same protocols have been followed for all the flights.

## 2.2   Model and Simulations Details

We used the regional chemistry transport model, WRF-Chem in this study, to simulate the observed vertical profiles of BC

and to understand the causes behind the sharp layers of BC at the elevated altitudes. The WRF-Chem model was employed with horizontal grid spacing of 2 km with 70 vertical levels and 100 m vertical resolution around the layers 4-6 km, 8-9 km and 10-11 km. Such a high vertical resolution was chosen to resolve the BC peaks, if simulated by the model, accurately. Out of the remaining levels, 10 are located within the boundary layer (0-2 km). A relatively coarser resolution (500 m), is set for the remaining altitude bands i.e. 2-4 km, 6-8 km and 9-10 km. Beyond 11 km, the vertical resolution is set to be

km. The details about the vertical levels in the model simulations can be found in table 1. The model domain (75.5$^0$E-80.5$^0$E, 14.5$^0$N- 20.5$^0$N) centers at Hyderabad and spans around 330 km in each direction. The simulations were started 16 days prior to the balloon flight days (flight day 1: 17$^{th}$ March, 2010, flight 2 : 8$^{th}$ January 2011 and flight day 3: 25$^{th}$ April 2011) and were run till the end of the flight day. In these simulations, the convective processes were not parameterized due to
the fine grid spacing; cloud microphysical processes were parameterized using the Thompson scheme (Thompson et al., 2004), boundary layer processes were parameterized using the YSU scheme (Hong et al., 2006; Hong, 2010), and the surface processes were modeled using RUC-LSM scheme (Smirnova et al., 1997, 2000). The short-wave radiation processes were parameterized using Dudhia scheme (Dudhia, 1989) while the long-wave processes were parameterized using RRTM (Mlawer et al., 1997). Gas-phase chemistry in these simulations was handled by MOZART mechanisms (Emmons et al., 2010), while the aerosol-
phase chemistry was parameterized using GOCART bulk aerosol scheme (Chin et al., 2002) with Fast-J scheme for photolysis (Wild et al., 2000).

The model takes into account the following aerosols species: BC1 (Hydrophobic), BC2 (Hydrophilic), OC1 (Hydrophobic), OC2 (Hydrophilic), dust (5 bins: effective diameters from 0.5 to 8 $\mu$m), sea-salts (4 bins: effective diameters from 0.1 to 7.5 $\mu$m) and sulphate. The characteristic conversion e-folding lifetime for BC from hydrophobic to hydrophilic i.e. BC1 to BC2 is
considered to be 2.5 days. More details about the treatment of BC in WRF-Chem can be found Kumar et al. (2015). The model simulates aerosol transport processes like emissions, advection, diffusion and deposition (dry and wet). The aerosol direct effects are modeled by coupling the aerosol scheme with the radiation scheme. The NCEP FNL (Final) Operational Global Analysis data, (interpolated to model resolution) has been used for setting up the initial and boundary conditions (updated every 6 h) for meteorological variables in the model, while the global chemistry transport model MOZART-4 (Emmons et al.,
2010) has been used for creation of initial and boundary conditions for chemistry variables. The near-surface emissions of gas-phase as well as aerosol-phase species are formulated using the standard emission pre-processor software PREP-CHEM-SRC (version 1) (Freitas et al., 2011). The RETRO database (Schultz et al., 2007) is used for various greenhouse and precursor gases, EDGAR (Olivier et al., 1996) for emissions of CO, NO, NH$_3$ and VOCs and the GOCART (Chin et al., 2002) database is used for the emissions OC, BC and SO$_2$ over the region. The necessary modifications in BC emissions from GOCART
database over this region (Govardhan et al., 2016) have been incorporated. We conducted two sets of simulations for each of the events. The first set included only surface emissions (NoACEM/Ctrl (Control) run: simulations without the prescription of AirCraft EMissions of BC) and the second set (ACEM- simulations with the prescription of AirCraft EMissions of BC), included surface and elevated emissions. The details of elevated emissions (BC emissions from aircrafts) are discussed in section 2.3)

**2.3 Prescription of aircraft BC emissions**

To understand the role of BC emissions from aircrafts in formation of the sharp layers of BC at elevated altitudes, we prescribed aircraft BC emissions from the MACCity global anthropogenic emissions inventory (Lamarque et al., 2010) in the WRF-Chem simulations. The inventory provides aircraft BC emissions at 0.5$^0$ × 0.5$^0$ grids and 23 vertical levels from 0.305 km to 13.725 km, with a vertical resolution of 610 m. The vertical profile of BC emissions from aircrafts, averaged over a 1$^0$ × 1$^0$ grid box

centering Hyderabad (fig.1, red line) and that over another grid box (fig.1, blue line) of the same dimension just $2^0$ south-west of the Hyderabad grid box, are shown in fig.1. The vertical locations of the emission peaks is seen to vary depending upon the proximity to airport (Hyderabad, in this case). The near-surface peak could be related to Landing-Take-Off (LTO) emissions over the airport, while the upper level peaks could be due to the emissions from the planes which are over-passing the location

without landing.

     The basic information this inventory considers are: global aircraft movements, performance characteristics of the different aircraft types, the actual paths of aircrafts during their journey and the emission factor for BC per kg of aviation fuel burnt. This global inventory for aircraft BC emissions inherently have several uncertainties, especially over a region like India mainly due

to uncertainties associated with a) amount of total aircraft fuel used across the country by domestic and international fleets; b) quality of aircraft fuel; c) age of aircrafts and hence the degradation in performance; d) exact number of civil aircrafts flying over the region e) unaccounted military aircrafts and f) exact routes followed by the aircrafts over this region. Additionally, a major source of uncertainty in the estimates of BC emission from aircrafts arises due to the uncertainties in the estimations of emission factor for BC 'EI(BC)' for the particular aircraft fuel being used. EI(BC) is the amount of BC emitted (in g) per kg

of aircraft fuel burnt and it depends on engine type, load conditions, engine conditions etc.. The measurement of EI(BC) by several studies (Herndon et al., 2008; Onasch et al., 2009; Timko et al., 2010) indicate that EI(BC) spans around four orders of magnitude. Stettler et al. (2013) discuss and quantify the underestimation of EI(BC) observed in currently available aircraft BC emissions inventories. In a previous study (Stettler et al., 2011) have stated: "The First-Order Approximation (FOA3) - currently the standard approach used to estimate particulate matter emissions from aircraft - is compared to measurements and

it is shown that there are discrepancies greater than an order of magnitude for 40% of cases for both organic carbon and black carbon emissions indices".

     Thus, to assess this inventory over Hyderabad (our study region), we estimated BC emissions from aircrafts over this region using previously reported values for fuel efficiency of different airline carriers (Kwan and Rutherford, 2015), seating capacity of the carriers (websites of different airlines), density of aviation fuel used (Cookson and Smith, 1990; Arkoudeas et al., 2003;

Outcalt et al., 2009; Blakey et al., 2011) in Indian region, BC emission index for the aviation fuel (Stettler et al., 2013), and actual data on air traffic over Hyderabad obtained from Air Traffic Controller, Hyderabad (Babu et al., 2011). Depending upon such evaluation of the MACCity emissions database over Hyderabad region, we have modified the inventory emissions by a corresponding scaling factor.

     Additionally, acknowledging the coarse resolution of the MACCity inventory, the 'line-source' nature of the freshly emitted

aircraft trail and the finer resolution of our model simulations, we confine the aircraft BC emissions into a width of 2 km and a height of 100 m. The mass conservation of the emitted BC due to such confinement leads to an additional scaling factor. The total scaling factor becomes the product of these two scaling factors. The modified emissions are formulated by multiplying the original emissions by the aforementioned scaling factors.

## 2.4 Computation of atmospheric heating rate

The atmospheric heating rates due to the vertical profiles of BC measured as well as simulated, are computed using the Discrete Ordinate Radiative Transfer Model (Santa Barbara DISORT Atmospheric Radiative Transfer - SBDART (Ricchiazzi et al., 1998)). SBDART solves radiative transfer equations considering a plane-parallel atmosphere. Aerosol are specified in the model through total aerosol optical depth, single scattering albedo (SSA) and the Legendre moments of the scattering phase function for the assumed aerosol mixture; at every vertical layer. To compute the heating rate associated with the observed BC profiles, the required layer-wise total AODs were computed using vertical profile total extinction coefficient from CALIOP (Cloud Aerosol Lidar with Orthogonal Polarization) LIDAR on-board the CALIPSO (Cloud Aerosol Lidar Pathfinder Satellite Observation) satellite. The level 2 extinction coefficient data from CALIPSO was cloud-screened using the standard, recommended cloud-screening algorithm (https://www-calipso.larc.nasa.gov/resources/calipso_users_guide/ tools/index.php) which makes use of the flags like AVD (Atmospheric Volume Description), CAD-Score (Cloud Aerosol Detection- Flag), extinction coefficient uncertainty and extinction coefficient quality control to separate aerosol from clouds. The final vertical profile of extinction coefficient considered was the mean all such cloud-screened vertical profiles which fall within $\pm$ 1.5$^0$ of the balloon flight location in x-y direction and also $\pm$ 20 days of the balloon flight date. Such an averaging was done to get a mean picture of the aerosol loading over the balloon flight region. To get the required single scattering albedo we made use of the observed vertical profile of BC and the OPAC (Optical Properties of Aerosol And Clouds) model (Hess et al., 1998). OPAC provides optical properties like extinction coefficient, absorption coefficient, scattering coefficient, single scattering albedo, asymmetry parameter, phase function etc. for the prescribed mixture of aerosol species. In OPAC, we prescribed the measured vertical distribution of BC to get the corresponding absorption coefficients. These absorption coefficients along with extinction coefficients from CALIPSO were used to get SSA. Finally, in order to compute the Legendre moments of the scattering phase function, we execute OPAC with one of its preset aerosol mixture- 'urban'; as it is best-suited for our observational location . This aerosol mixture has water soluble species (28000 # cm$^{-3}$), insoluble species (1.5 # cm$^{-3}$) and soot (130000 # cm$^{-3}$). For the 'urban' aerosol mixture we derived the scattering phase function and computed 8 Legendre moments of the scattering phase function. Along with the aforementioned primary inputs, SBDART requires some additional inputs such as mean state of atmospheric variables viz. temperature and pressure on every vertical level; and vertical profile of specific humidity and ozone. These inputs were taken from SBDART database under 'tropical' category, which suits our observation location (Hyderabad). Additionally, spectrally varying surface albedo values for Hyderabad were taken from MODIS surface reflectance product (MOD09CMG). This daily level 3 product provides surface reflectance for 7 bands (469 nm, 555 nm, 645 nm, 858.8 nm, 1240 nm, 1640 nm and 2130 nm) at 0.05$^0$ resolution. With all these inputs SBDART model was executed with 8 streams in the radiative transfer calculations with 7.5$^0$ resolution for solar zenith angle varying from 0 to 180. The net radiative fluxes were computed at top and bottom of every vertical layer starting from surface to 100 km. A similar run of SBDART was conducted without the prescription of aerosols. Using radiative fluxes from these 2 runs, radiative forcing within each layer was estimated using

$$\Delta F = Flux_{NA} - Flux_A \tag{1}$$

where, $\text{Flux}_{NA}$ = Radiative flux in absence of aerosols (W m$^{-2}$)

$\text{Flux}_A$ = Radiative flux in presence of aerosols (W m$^{-2}$)

The change in radiative fluxes in each layer due to aerosol is the amount of energy absorbed in the layer due to aerosols. Corresponding atmospheric heating rates were computed using Liou (2002),

$$dT/dt = (g/Cp)(\Delta F/\Delta p) \tag{2}$$

where dT/dt is the heating rate (K s$^{-1}$), g is the acceleration due to gravity, Cp the specific heat capacity of air at constant pressure (Cp=1006 J kg$^{-1}$ k$^{-1}$) and $p$ is the atmospheric pressure (Satheesh and Ramanathan, 2000). The heating rates (K day$^{-1}$) were then averaged for 24 hours.

The vertical extents of heating rate profiles from the measurements were limited due to the availability of extinction coefficient data from CALIPSO over the region of interest. Such a limitation does not exist for model data, hence we have computed the atmospheric heating rates using the model results. To compute the atmospheric heating rates using model data we follow exactly the same aforementioned procedure. The only differences are: we use extinction coefficient data from model simulations instead of the CALIPSO satellite and we use SSA from model simulations instead of deriving it from the observed BC.

## 3   Results

### 3.1   Observed vertical profiles of BC

The vertical profiles of BC, derived from flights measurements are shown in fig.2a-c, in which each profile is the average of the ascent and descent profiles for that flight. All the profiles revealed sharp and confined peaks in BC concentration in the free troposphere above a near-steady mass concentration within the planetary boundary layer ($\sim$ 2 km for pre-monsoon and $\sim$ 1km for winter). These peaks are identified on the respective profiles in fig.2a-c. It is also seen very clearly that the sharp and most prominent peak occurred at between 4 and 5 km in the first and last profiles (fig.2a and fig.2c); which incidentally correspond to the pre-monsoon flights, while it occurred at a slightly lower altitude (between 2 to 3 km) in the winter profile. The highest altitude layer seen in the profiles were in the vicinity of 7-9 km, though the amplitudes of these are much smaller than that of the layer seen at around 4.5 km. With the help of temperature data from the collocated meteorological payloads, we evaluated the environmental lapse rate profile for each of these profiles and the mean profile for each flight is shown in fig.2d-f, following the same order as in fig.2a-c. It is clearly seen that in the vicinity of the prominent BC peaks (P1 to P5 in fig.2a-c) there is a large reduction in the environmental lapse rate (sometimes reaching close to zero) e.g. at 4.5 km during March 2010 (fig.2d), around 2-3 km and 7 km during January 2011 (fig.2e); and around 4-5 km and 6-7 km during April 2011 (fig.2f). The lapse rate profile during April 2011 (fig.2f) shows multiple spikes possibly due to higher number of peaks in the corresponding BC profile (fig.2c), (Since the raw temperature data is very noisy, we have smoothened the data using two points fixed and two points moving average filter). Such reductions in temperature lapse rate would affect the local stability scenario. While the

occurrence of high BC below 3 km can be explained by the boundary layer dynamics, high BC values at higher levels especially at 7 to 9 km, remained a mystery. Babu et al. (2011) hypothesized that the occurrence of such BC peaks at high altitudes could be associated with the local sources of BC at those altitudes, as 4-5 km and 8-9 km are the preferred corridors for aircrafts flying over Hyderabad. With a motivation to test this hypothesis and also to estimate the climate implication of such BC layers, we in

this study, investigate the causes for the occurrence of such sharp and confined BC peaks at higher altitudes, using a regional online chemistry transport model. Crucial findings from this study are reported and possible ramifications are discussed.

## 3.2   Meteorological evaluation of the model

Meteorological processes like advection, diffusion, deposition etc. play a significant role in controlling the concentration of pollutants near the surface and their vertical distribution in the troposphere (Govardhan et al., 2015). Hence, before inspecting

the model simulated vertical profile of BC, we first examine the performance of the model in simulating meteorology over the region of interest. We show such comparisons for March 2010 NoACEM simulations as a representative one. The model simulated meteorological parameters are averaged over the balloon flight region and then compared with the corresponding observations from the balloon flight. The three-point running mean smoothing is used to smoothen observational and simulated vertical profiles. Firstly, we have compared the model simulated vertical profile of horizontal wind speed (Blue line, fig.3 a)

and direction (Blue line, fig. 3b) with the corresponding observations (Red line, fig.3a and Red line, fig.3b) available from the balloon flight measurements (deduced from the GPS data on board the balloon). The figures depict relatively weak (less than 10 m s$^{-1}$) low-level northerly-northeasterly winds upto around 5 km. Beyond this altitude, the winds change direction gradually and become westerlies above 8 km. The wind speeds also increase drastically beyond 8 km and reach a value of 20-25 m s$^{-1}$ at a height of 9 km and beyond. While the model simulated vertical variation of wind speed (Blue line, fig.3a) and direction (Blue

line, fig.3b) agree broadly with the measurements, they differ in details and magnitudes. In model simulations, the change in wind direction starts occurring at an altitude $\sim$ 1 km lower than in observations. The large increase in wind speed beyond 8 km altitude is satisfactorily captured by the model. Thus, with some disagreements in the actual magnitudes, the model captures broad features of horizontal advection strength (wind speed) and nature (wind direction) over the balloon flight domain vis-a-vis the measurements. We next examine the model simulations of vertical stability over the flight domain by comparing

the model-simulated vertical profiles of potential temperature ($\theta$) with the corresponding profiles from the balloon data. The observed profile (Red line, fig.3c) shows a stable layer ($\theta$ increases with height) up to first 2 km of the lower troposphere. Above this layer, there occurs a thick, well-mixed layer upto a height of 4 km, in which $\theta$ remains almost height-invariant. Above this height, the atmosphere remains largely stable with increase in $\theta$ with height. The modeled altitude variation of $\theta$ (Blue line, fig.3c) in general, matches with the measurements, barring a few discrepancies such as the extent of stable and well

mixed layer in the lower atmosphere, magnitude of $\theta$ in the vicinity of the primary BC maxima in observations (around 4-4.5 km, fig.2a). The model captures the stable layer lying up to around 1.5 km, similar to that seen in observations (up to 2 km, red line, fig.3c). A convectively unstable, well-mixed layer ($\theta$ constant) extending upto a height of 3.5 km, occurs above this layer, akin to observations (up to 4 km, red line, fig.3c), with an underestimation in $\theta$ values. Beyond this height, model also does not show any sign of instability, while $\theta$ increases with height, in agreement with the observations, yet with differences in the

magnitudes. Thus, the model simulated vertical profile of $\theta$ also appears to be broadly comparable to the observed profile, with some differences. Examining the vertical thermal structure, we have also compared the model simulations of vertical profile of temperature with the corresponding measurements from the flight. While showing in general reduction in temperature with height similar to the observations (Red line, fig.3d), the model (Blue line, fig.3d) shows discrepancies in actual magnitudes vis-a-vis the observations (-1.7 to +3.4 K), as in case of the other meteorological variables. Possibly, owing to the existence of the primary BC peak between 4 to 5 km, the observations (Red line, fig.3d) depict a near-steady temperature within the particular altitude layer. The model (Blue line, fig.3d) on the other hand, fails to capture this feature. Thus, along with some differences in the details, the model simulations capture some of the large-scale features of the meteorology over the balloon flight region vis-a-vis the observations.

## 3.3   Simulated vertical profile of BC

With the above broad agreements in the meteorological fields, we proceeded to examine and evaluate the model simulated vertical profile of BC in the vicinity of the balloon flight domain (blue line, fig.2g-i) in the NoACEM configuration. For $17^{th}$ March 2010, the model simulations (NoACEM) show considerable differences vis-a-vis the observed vertical profile of BC (fig.2a). The magnitudes of the simulated BC (blue line, fig.2g) show a relatively good comparison with the observations only over the lower most altitudes i.e. below 3.5 km. Above that altitude, the 2 profiles completely differ from each other; while the observations (fig.2a) show an increase in BC followed by a sharp peak at 4.5 km (BC $> 12$ $\mu$g m$^{-3}$), the model (fig.2g) displays a rapid decrease in BC concentration with altitude, without even a sign of the 'elevated' (high altitude) layers of high BC concentration seen in the measurements. Thus, though the model satisfactorily simulates the meteorology over the flight domain, it does not simulate the observed vertical profile of BC. The results for the other two flights are shown in fig.2h and fig.2i. During January 2011, the model simulations (NoACEM) (blue line, fig.2h) show rapid reduction in BC within first 1 km. There is no sign of sharp and confined peaks beyond this height unlike the corresponding observations (fig.2b). During April 2011, the model simulations (NoACEM) (blue line, fig.2i) show sharp reduction in BC within first 1 km. Beyond this height, a sharp peak in BC is seen with a maxima near 2 km. A relatively gradual reduction in BC occurs beyond this height upto 4 km, followed by a flat steady BC profile with very low values above that height (blue line, fig.2i). Thus, none of the model simulations capture the sharp-confined BC peaks (or the elevated BC layers) occurring at higher altitudes. This strongly suggests that the meteorological factors alone cannot be responsible for the existence of the elevated BC layers. To throw more light on this, we considered other hypothesis.

The meteorology (observed as well as simulated) being benign for all the cases, and possibility of any long-range transport of BC from other location being an unlikely cause for such high concentrations (Babu et al., 2011), one has to look for local injection of BC aerosols at mid and upper troposphere because, surface based emissions would not lead to elevated BC layers at altitudes of 4-5 km or 7-9 km. In this context, emissions from commercial air traffic (that overfly Hyderabad, as well land and take off from there) assume significance. In their first reporting of the elevated layers, Babu et al. (2011) have hypothesized the role of such emissions, where they have obtained an approximate estimate of air traffic over the study location; about 200 aircrafts that overfly Hyderabad in the corridor 8-10 km and another 250 to 300 km that use the corridor 4 to 5 km, in the

course of landing and taking off from the airport. We next evaluate elevated emissions due to aircrafts as a likely candidate for these high concentrations.

### 3.4 Simulated vertical profile of BC with aircraft BC emissions

We examine the outcome of prescription of BC emissions from aircrafts on the modeled vertical profile of BC in fig.2g-i, in the vicinity of the balloon flight domain, for the 3 balloon flight days. It is quite interesting to note that, upon the prescription of BC emissions from aircrafts, the model simulations (ACEM- simulations with the prescription of AirCraft EMissions) show sharp layers in the vertical profiles akin to the observed BC profiles (red line, fig.2g-i), which are not simulated otherwise (NoACEM/Ctrl (control) run- simulations without the prescription of AirCraft EMissions). Though the actual altitudes and the magnitude of the BC layers differ in comparison with the observations, the sharpness of the modeled BC layers make them look similar to the observed BC layers. The two peaks in BC profile during March (red line, fig.2g), the lower and the upper level BC peaks during January (red line, fig.2h) and the clustered BC peaks during April (red line, fig.2i) are well simulated by the model only after the prescription of BC emissions from aircrafts. Thus, even for the model with realistic meteorology, the high altitude BC peaks/ layers are captured only when the high altitude sources of BC are prescribed. This clearly highlights the role played by aircraft emissions of BC in creation of the high altitude BC peaks, which remained as a hypothesis earlier. Additionally, as a consequence of such high altitude BC emissions, the modeled vertical profile of temperature lapse rate over the balloon flight region shows a reduction in the magnitude in ACEM case as compared to the NoACEM case, indicating the warming due to BC (fig.4a). The corresponding differences in the temperature lapse rate ($d(dT/dz)= (dT/dz)_{ACEM}$ - $(dT/dz)_{NoACEM}$) appear to be higher especially at higher altitudes of 4 km and beyond 7km (fig.4b). Such a reduction in the magnitude of the temperature lapse rate values, results in a better match between model simulations and the corresponding observations of temperature lapse rate especially at higher altitudes (from 6 to 8 km and from 9 to 9.5 km)(fig.4a).

Additionally, we carried out one more model simulation, in which we prescribed the emissions of BC from biomass burning activities using the Fire INventory from NCAR (FINN) version 1.5 inventory (Wiedinmyer et al., 2011) biomass burning data. The FINN provides high resolution, global emission estimates from such open burning activities. The temporal resolution of the inventory is 1 day, while the spatial resolution is 1 km$^2$. In our simulations, we allowed such emissions from biomass burning to lift vertically following the online plume-rise module (Freitas, 2007). However, the model could not simulate the observed sharp and confined BC peaks (figure not shown), if we switch off the prescription of BC emissions from aircrafts. This clearly shows the important role of aircraft BC emissions in causing high altitude BC peaks.

### 3.5 Sensitivity of the vertical profile of BC to the surface level emissions

To test the robustness of the simulated high altitude BC peaks, we have also examined the impact of near-surface emissions of BC on the elevated layers. For this, we have done a similar simulation with aircraft BC emissions, in which we have turned-off the near-surface fossil fuel emissions of BC over the model domain. We have then examined the effect of this on the simulated vertical profile of BC in fig.5. It can be clearly seen from this figure that the vertical profiles simulated by the 2 runs (SE- the run with prescribed near-surface anthropogenic emissions of BC and N-SE - the run with near-surface anthropogenic BC

emissions turned off) differ only in the lower altitude region, upto a height about 4 km (probably the altitude up to which BC could be lofted by the convection). Beyond 4 km, the profiles are largely similar, implying that the elevated BC layers are insensitive to surface BC emissions. Beyond 4 km, the profiles look similar. The correlation coefficient between the two BC profiles beyond 4 km comes out to be 0.97 which is 99.99% significant. Moreover, the magnitudes of BC in the two profiles

show good agreement with a difference limited to only 0.1 $\mu$g m$^{-3}$. Thus, our model simulations indicate that, the high altitude peaks in BC are not a result of convective-lifting of near-surface BC, but they are indeed caused due to the BC emissions (injection of BC) at higher altitudes by the aircrafts.

## 3.6   Model simulated elevated sharp BC layers: Seasonal Scenario

We noticed that the model, WRF-Chem, produces the elevated sharp peaks of BC akin to the observations upon the prescription

of aircraft BC emissions from the regionally fine-tuned MACCity inventory. We now proceed to examine the seasonal behavior such elevated sharp BC layers, within the model simulations. For this, we have carried out 1 month long model simulations during each season of the year 2010 i.e. January, March, July and October, representative of winter, pre-monsoon, monsoon and post-monsoon seasons. Keeping in mind, the limited width and the dynamic behavior of the horizontal location of the aircraft emitted trail, instead of plotting a monthly mean vertical profile of BC, averaged over a region, we have computed

seasonal probability of occurrence of a high altitude BC peak at every point within the domain.

### 3.6.1   Probability of occurrence of sharp BC peak within 9-11 km

For every point within the domain, the probability of occurrence of sharp BC peak is computed by examining the maximum value of the simulated BC within 9-11 km altitude band and that within 7-9 km altitude band. When the ratio of maximum

BC magnitude within 9-11 km band to that within 7-9 km band, is more than 3, we considered the occurrence of high altitude sharp BC peak within 9-11 km altitude band. The probability of occurrence of sharp BC peak within 9-11 km band for the entire month is computed as follows,

$$p_{(9-11)} = \frac{T_{(Peak:9-11)}}{T_{Total}} \tag{3}$$

where,

p$_{(9-11)}$ = probability of occurrence of a sharp BC peak within 9-11 km altitude band

T$_{(Peak:9-11)}$ = Number of instances when a sharp BC peak occurs at 9-11 km altitude band

T$_{Total}$= 240 for the entire month with 3 hourly output frequency

Such computation of p$_{(9-11)}$ is done for all the months of the model simulations and is shown in fig.6. During the month

of January 2010 (fig.6a) and March 2010 (fig.6b), the probability of getting an upper level (9-11 km) sharp peak in BC within the model domain goes as high as 60% over the area in the south-west part of the domain. This looks to be primarily related to the location of the aircraft emitted trail in the emissions inventory. Around 40% chances of getting a sharp peak at upper level

(9-11 km) are seen at various locations which roughly occur over the location of the aircraft emissions in the inventory and are further controlled by the direction of the prevailing upper level winds. For the monsoonal period i.e. during the month of July 2010 (fig.6c), such probability values reduce, with most of the region showing less than 10% probability. Such a reduction in $p_{(9-11)}$ over the domain looks to be a result of convective lifting of BC during monsoon (which is relatively lesser during pre-monsoon and post-monsoon) resulting into more uniform concentration of BC at the elevated altitudes causing removal of such sharp peaks in BC, but increased abundance of BC at those altitudes as shown in fig.7. During the month of October 2010 (fig.6d), the $p_{(9-11)}$ once again reaches to as high as 50%, with the specific aircraft emission zones showing higher values. The spatial pattern of the probability appears to be roughly contrast of that during January or March 2010. This behavior could be due to the difference between the direction of the prevailing winds during January/March and that during October. Thus, in summary, probability of getting a BC peak at 9-11 km altitude in the model simulations, looks to be dependent on the season and the location.

### 3.6.2  Probability of occurrence of sharp BC peak within 4-5 km

Following a similar exercise, we have estimated the probability of occurrence of BC peak within 4-5 km altitude band. The lower level (4-5 km) peak in BC is identified when the maxima of BC magnitude within the 4-5 km altitude band goes more than twice of that of the neighboring altitude bands. The spatial plot of such probability values for each month is shown in fig.8.

In-general, it could be seen that the probability of getting a lower level (here 4-5 km) peak in the simulated BC (fig.8) is lower than that for the upper-level (9-11 km) peak (fig.6), within the model simulations. This could possibly be linked with the differences associated with, the altitude variation and the magnitudes, of BC emissions from aircrafts, over 9-11 km altitude band vis-a-vis 4-5 km altitude band, within the inventory. During July 2010, the monsoonal convection reduces the $p_{4-5}$ values by vertically lifting BC throughout the column, getting rid of the sharp BC peaks as shown in fig.7.

### 3.7  Atmospheric heating rate due to the observed and modeled BC vertical profile

Being a strong absorber of radiation over a wide spectral band, elevated BC layers absorb incoming solar, which would then heat-up the atmosphere locally and can alter the vertical stability. In this section, we compute the atmospheric heating caused by the observed and modeled BC vertical profiles. The methodology that we follow to compute the atmospheric heating rates due to BC is similar to that followed by Babu et al. (2002) and Babu et al. (2011).

### 3.7.1  Atmospheric heating rates due to observed BC

Following the procedure outlined in section 2.4, we computed the atmospheric heating rates due to the measured BC profiles for the 3 balloon flights. The heating rate profiles are presented in fig.9a, fig.9b and fig.9c for the March, January and April flights respectively. The vertical extent of the profiles is limited to the availability of the extinction coefficient data from CALIPSO

satellite. During March 2010 (fig.9a), the heating rates are seen to be within 0.5 to 1 K day$^{-1}$ upto 2-2.5 km of the atmosphere. The denser atmosphere could also be one of the factors responsible for such low values of heating rates at lower altitudes. Beyond this height, a large increase in heating rates could be seen up to ∼5 km. The heating rate maxima (∼5 K day$^{-1}$) is seen to occur around the height of BC maxima (fig.2a). Such a large heating rate could cause the observed reduction in temperature lapse rate around those altitudes (fig.2d). During the winter flight (January 2011, fig.9b), the observed BC profiles along with extinction coefficient profiles from CALIPSO, cause more heating near the surface. The corresponding heating rates at near-surface levels are even higher than that during the summer months (March 2010, fig.9a and April 2011, fig.9c). The heating rate profile during April 2011 (fig.9c) shows similar features as that of March 2010 (fig.9a). The maxima in heating rate also occurs around the elevated BC layers, with comparable magnitudes. As expected, since the observed BC profile during April 2011 (fig.2c) shows more number local peaks as compared to that during March 2010 (fig.2a), the heating rate profile during April 2011 also shows more pronounced spikes vis-a-vis March 2010. The largely similar nature of the heating rates profiles during March 2010 and April 2011, brings out the average features of heating rate profile during summer months over the region of study. Thus, during the summer months the study region is characterized with maxima in the heating rates (∼5 K day$^{-1}$) at around 4-5 km, while during the winter months the maxima (∼1 K day$^{-1}$) lies near the surface.

### 3.7.2  Atmospheric heating rates due to modeled BC

The corresponding atmospheric heating rate profiles in model simulations are shown respectively in fig.9d,e and f. The heating rates are computed for the corresponding BC profiles (ACEM) shown in fig.2g,h and i (red line). The profiles are representative of all such profiles, which show the effect of aircraft emissions. During March 2010, the model heating-rate profile (fig.9d) shows similar features vis-a-vis the corresponding observations (fig.9a). Though the actual magnitudes of heating rates are lower for the model simulated profiles, it captures the major and important features such as steady increase in heating rates from surface to 3 km and maxima in heating rates at around 3-4 km. Interestingly, a secondary peak in heating rates can be seen around 11 km. This peak corresponds to the peak in BC as seen in fig.2a. During January 2011 (fig.9e), the model also shows high values of heating rates nearer to the surface similar to the observations (fig.9b). Beyond this height the heating rates reduce. A secondary peak in heating rates at around 10-12 km is seen also during January 2011, with the magnitude as high as 0.5 K day$^{-1}$. During April 2011, the model heating rate profile (fig.9f) shows a maximum close to the surface corresponding to the maximum in BC (fig.2c), and multiple maxima at higher heights (3km, 6km) corresponding to the presence of BC layers. The upper level maxima at around 11 km is seen to be relatively weaker during April (fig.9f) vis-a-vis March (fig.9d) and January (fig.9e) simulations. The heating rate profiles shown in fig.9d-f, are a representative of all those locations which show an existence of the aircraft emitted BC trails at higher altitudes, but are located away from the area with high emissions (the region, with high magnitude of emissions compared to the rest of the trail, where two or more such trails cross each other). We term these generic heating rate profiles as 'normal profiles'. Additionally, we inspected the heating rate profiles for those locations within the trail which lie in the vicinity of the area with relatively high emission intensity. Such regions are formed when two or more such trails cross each other. We term the corresponding heating rate profiles as 'extreme profiles'. Such 'extreme profiles' of heating rates are shown in fig.9g-i, for March 2010, January 2011 and April 2011 respectively. It

can be noticed from fig.9d-i that, while the extreme profiles show similar heating rate magnitudes within lower part of the atmosphere as that of the normal profiles; they produce very high heating rates at the higher heights (10-12 km). Such heating rates even cross the near-surface maxima for the April 2011 profile (fig.9i). The rarer atmosphere could also contribute to such a high values of heating rates at those altitudes. Such large heating rate values at those heights can affect the stability of the atmosphere. The resulting vertical mixing can lift the BC particles even higher. This continuous warming-lifting cycle can result in further vertical transport of the BC particulates, which is termed as self-lifting of BC (de Laat et al., 2012). In addition to the self-lifting mechanism, the BC emitted at high altitudes can get transported vertically up in presence of strong underlying convection. In the next section, we show, with the help of model simulations, that such convective transport of the high-altitude BC could occur in tropics during periods of strong convection.

## 3.8 Convective lifting of BC

To examine the convective lifting of high-altitude BC, we have carried out WRF-Chem simulations with the prescription of aircraft BC emissions over Indian region for July 2010. July month was selected for this purpose, in view of the known prevalence of deep convection over Indian region, associated with the summer monsoon and the tropical tropopause layer being thinnest at this time of the year (Meenu et al., 2010; Sunilkumar et al., 2013). The animation (Mov.1 in supplementary material) shows height-longitude plots of maximum value of BC for a latitude belt of $2.25^0$ over Hyderabad (X-axis: Longitude, Y-axis: Height (m)). The animation clearly reveals the capability of strong convections to lift BC to heights beyond 14 km. On a few occasions, these layers are seen to get transported even beyond 17 km, across the tropopause. Similarly, we compute the difference in the upper tropospheric BC mass concentration values in ACEM (runs with the prescription of aircraft BC emissions) simulations vis-a-vis the NoACEM case (runs without prescription of aircraft BC emissions) ($\Delta BC = BC_{ACEM}$-$BC_{NoACEM}$), averaged for the entire duration of model simulations carried out during July 2010. The maximum of such time-averaged $\Delta BC$ values across the latitude belt of $2.25^0$ centered over Hyderabad, for every longitude of the model domain is shown in fig.10. The time-averaged increment in UTLS (Upper Troposphere and Lower Stratosphere) BC load due to aircraft BC emissions is seen to occur over the entire longitude belt, though more pronounced over a few longitude bands ($79^0E-80^0E$). Higher $\Delta BC$ values extend vertically even beyond 17.6 km, highlighting the mean vertical transport of aircraft emitted BC in the UTLS region. These provide a strong evidence for the intrusion of high altitude aircraft emitted BC into the upper-tropospheric-lower-stratospheric heights over the Indian region during strong convective periods. The intrusion of such tropospheric air into the stratosphere will be favored under certain conditions like thinner tropopause layer, unstable conditions within the tropopause layer etc. In the next section, we show observations supporting such intrusion.

### 3.9 Occurrence of BC at high altitudes: Observational evidence

There exist several observational reports of presence of BC at upper tropospheric and the stratospheric altitudes (Chuan and Woods, 1984; Okada et al., 1992; Pusechel et al., 1992; Sheridan et al., 1994; Blake and Kato, 1995; Pueschel et al., 1997; Strawa et al., 1999; Baumgardner et al., 2004; Schwarz et al., 2006; Kremser et al., 2016). One of the ways to ascertain the occurrence of such layers in the stratosphere is to examine the CALIOP LIDAR (on board CALIPSO satellite) extinction coefficient data (at

550 nm). We have examined all such vertical profiles for 3 consecutive years (from January 2010 to December 2012), over 5 specifically chosen regions ($5^0 \times 5^0$) over and around India (fig.11). Since, we want to focus on the presence of aerosols at stratospheric altitudes, we have plotted the vertical profiles of extinction coefficient in the stratospheric altitudes (i.e. from altitude 20 km to $\sim$ 30km). The results are shown in fig.12.

To our surprise, we notice that the aerosol extinction coefficient values over the stratospheric altitudes are more than 0.02 $km^{-1}$ over all these regions; occasionally values greater than 0.06 $km^{-1}$ are also noticeable (fig.12a). The corresponding appended time-series of stratospheric (i.e. from altitude 20 km to $\sim$ 30km) Aerosol Optical Depth over the regional boxes under consideration from 2010 to 2012 is plotted in fig.12b. The AODs are seen to be higher than 0.01 on most of the occasions with frequent spikes reaching as high as 0.16. We considered different scenarios to explain such high AOD. These included,

(i) volcanic perturbation to stratospheric AOD, and (ii) high altitude cirrus clouds at TTL region. We first compared these extinction coefficient and AOD values with the corresponding area-averaged values for the entire tropical belt ($20^0$S to $20^0$N) (Kremser et al., 2016). The maximum value of stratospheric AOD averaged for the entire tropical belt (Kremser et al., 2016) for the period 2010 to 2012 (which is considered to be the background tropical stratospheric AOD in this study), is marked with the dotted red-line in fig.12b. It can be seen that, the stratospheric AODs from our analysis over the 5 regional boxes are

roughly 4-6 times higher than the background tropical stratospheric AOD (dotted red line, fig.12b), same is the case for aerosol extinction coefficient values as well (comparison not shown). The large differences in the background tropical stratospheric AOD and the stratospheric AODs over the Indian region, signify the enhancement in the stratospheric aerosol burden over the regions under consideration, above the background stratospheric aerosol loading. One of the major controllers of stratospheric AOD is the in-flux of particulates namely sulphates, organics and ash, and gases like $SO_2$ from volcanic eruptions

into the stratosphere (Kremser et al., 2016). An examination of the occurrence of global volcanic eruptions from 2008 to 2012 (Kremser et al., 2016) suggests that, out of the 11 major volcanic eruptions that occurred during that period, 7 were centered away from the tropics. Also, the flux of $SO_2$ in the stratosphere from the most intense of the 11 events, was merely $(1/10)^{th}$ of that of the mount Pinatubo eruption (Kremser et al., 2016). Thus, this proposes that the enhancement in AOD and the extinction coefficient over the regions under consideration in this study may not be linked to volcanic eruptions. The monsoon season

(June to September) appears to be the favorite period for the lifted layers in the stratosphere (fig.12a). This looks to be linked to the vertical lifting associated with severe convection (as seen in our model simulations fig.10 and Mov.1 in supplementary material) and thin tropopause layer as explained earlier (Fu et al., 2006; Randel et al., 2010; Vernier et al., 2011; Thampi et al., 2012; Uma et al., 2012). Additionally, these layers seem to be present during the other seasons (pre-monsoon and winter) as well. The Indian region experiences a large scale ascending motion above 10 km during winter months (Rao et al., 2008). Such

an ascending motion can cause intrusion of tropospheric air (lying at 8-10 km) into the stratosphere (Rao et al., 2008). Thus, while the monsoonal convection and the related thermodynamics appears to be a cause for the summer time higher extinction coefficient values at stratospheric heights over the Indian region, the transport of tropospheric air mass into the stratosphere due to the large-scale ascent beyond 10 km looks to be responsible for the winter time high values.

    To throw light on the aerosol species responsible for such values of extinction coefficient, we examined the Particle Depo-

larization Ratios (at 550nm, henceforth PDR) at these altitudes (fig.12c). PDR is the ratio of attenuated backscatter coefficient

in perpendicular direction to that in the parallel direction. We computed the PDR values from the corresponding backscatter coefficient values. The PDR values are seen to be higher than 0.3 on most of the occasions. This suggests the presence of non-spherical aerosol species at those altitudes. One of the major non-spherical aerosol species over this region is mineral dust. The source for mineral dust aerosol species in stratosphere could be largely related to a) volcanic eruptions b) meteoritic debris

and c) convective transport of tropospheric dust. As mentioned previously, our study has been carried out for a period which is relatively volcanically quiescent. Thus, volcanically erupted dust may not have contributed much to the stratospheric aerosol burden during our study period. The meteoritic debris form a minor part (5-10%) of the stratospheric aerosol composition, especially at altitude less than 30 km (Turco et al., 1981), making them difficult to be captured by a LIDAR. Moreover, the meteoritic debris are reported to be coarse in size (i.e. having radius above 1 $\mu$m) (Turco et al., 1981; Mackinnon et al., 1982)

(and hence would have large deposition rates) and are largely spherical in shape (Mackinnon et al., 1982) with lower values of particle depolarisation ratio (PDR less than 0.1 (Klekociuk et al., 2005)). The extinction coefficient associated with plume of meteoritic debris are 3 orders of magnitude lesser (Gorkavyi et al., 2013) than the values we notice over our study domain. These points together rule out the possibility of associating the observed values of extinction coefficient and PDR to the meteoritic dust. The convectively lifted dust and other air pollutants while can get into the higher altitude regime (Andreae et al.,

2001; Randel et al., 2010; Vernier et al., 2011; Fadnavis et al., 2013; Corr et al., 2016), their transport over the Indian region is limited to around 20 km (Fadnavis et al., 2013), which is mainly governed by the height of convective towers (Meenu et al., 2010). Moreover, dust aerosol is 2.6 times heavier than BC (Hess et al., 1998) and also less solar absorbent; thus it is unlikely to get vertically lifted up beyond 20 km by large-scale or self lifting mechanisms (de Laat et al., 2012). Hence we eliminate the possibility of existence of tropospheric dust as well beyond 22 km altitude over our domain. Additionally, the maximum cloud

top altitude occurring over the Indian region during the monsoon season is 20 km (Meenu et al., 2010). This suggests that, the non-spherical ice crystals emanating from the anvils of the convective towers over the Indian region could be found only around 20 km and not above 21-22 km. These together eliminate the possibility of presence of non-spherical mineral dust particles and ice crystals at these heights. A few previous studies have noticed presence of BC chain agglomerates at the upper tropospheric and lower stratospheric heights (Blake and Kato, 1995; Bekki, 1997). Such a transport of BC to the stratospheric heights could

be a result of convective-lifting (as seen in our model simulations fig.10 and Mov.1) and 'self-lifting' (de Laat et al., 2012) associated with them. Though pure and nascent BC has relatively lower value of PDR, the aged BC particles being porous (Chand et al., 2005; Dusek et al., 2005) and capable of forming long chain agglomerates are non-spherical and would depict higher values of PDR. Such BC agglomerates even when mixed with stratospheric sulfate could still give rise to non-spherical shapes and thus higher values of PDR. This indicates the possible existence of BC at the stratospheric altitudes over the Indian

region. Thus, the model simulated convective transport of high altitude aircraft emitted BC layers to UTLS region, favorable conditions for cross-tropopause transport of air-mass over Indian region during monsoonal months, large scale winter-time ascending motion above 10 km over Indian region and the presence of local BC emission sources (aircrafts) at high altitudes ($\sim$ 10km) throughout the year together indicate the possibility of aircrafts emissions being primarily responsible for the possible presence of BC in stratosphere over the Indian region.

## 4 Possible implications of presence of BC at stratospheric heights

Our model simulations suggest that, aircrafts emissions are capable of producing the observed elevated aerosol layers in free troposphere (middle and upper) and the BC particles in these layers can intrude into the stratosphere, especially during periods of thin TTL, aided by strong convective lofting (associated with summer monsoon) and self-lifting mechanisms. The average residence time of stratospheric aerosol is about 1 year (Snetsinger et al., 1987). BC aerosols, owing to their porosity and fractal shape (Blake and Kato, 1995; Bekki, 1997), can provide larger surface area to support heterogeneous chemical reactions in the upper troposphere and lower stratosphere. One of such reactions is the ozone decomposition on BC (Akhter et al., 1985; Bekki, 1997; Schurath and Naumann, 1998; Disselkamp et al., 2000; Chughtai et al., 2003; Satheesh et al., 2013), which may result in depletion of stratospheric ozone. Besides, if BC is transported to the stratospheric heights, the high altitude cirrus clouds existing around the tropical tropopause layer, which reflect back most of the solar radiation incident on them, would consequently induce increased interaction between solar radiation and the lifted BC layer (Satheesh and Moorthy, 2005). This would substantially enhance BC-induced warming of lower stratosphere. Since the atmosphere becomes increasingly thinner with altitude, for a given amount of absorbed energy, the warming would be higher at upper levels than that near the surface. Kamm et al. (1999), based on laboratory experiments, have reported that reaction rate of loss of ozone in presence of BC has positive temperature dependence. Thus, the aircraft emitted BC could possibly contribute to the depletion of the stratospheric ozone layer. The continued negative trend in stratospheric ozone over $40^0$S-$40^0$N from 1984 to 2014 (Bourassa et al., 2014) could possibly be related to the ever increasing aircraft traffic (Paraschis and Gittens, 2014; Piccolo and Gittens, 2015, 2016). The observed delays in recovery of stratospheric ozone hole (Kane, 2008, 2009), could also be influenced by such interactions of lifted BC with the stratospheric ozone.

Thus, with the help of a regional chemistry transport model, our study showed that the aircrafts (with emissions from the regionally fine-tuned MACCity inventory) appear to be one of the primary causes behind the occurrence of the sharp elevated layers of BC over the Indian region. This study utilized high altitude measurements of BC carried out using a zero-pressure balloon. Though, the balloon measurements were carried out in 3 different seasons to capture the seasonal picture, they are limited in number. In future, more number of such high altitude balloon measurements of BC would be needed to confirm the occurrence of such elevated sharp BC peaks. Also, though the high altitude balloon launching facility is currently available only at 1 station in India, more number of such balloon measurements from different locations, using mobile facility would be useful to generalize the results. Additionally, such high altitude BC measurements could also be carried out in the vicinity of busier airport locations across the world to throw more light on such implications of aircraft emissions. The BC emissions from aircraft, in the MACCity inventory were scaled in this study, with necessary modifications. This highlights the large uncertainty associated with BC emissions from aircrafts, especially regarding the emission factor for BC (EI(BC)) and possibly with actual air traffic data. This uncertainty needs to be examined in future to constrain the implications of emissions from aircrafts.

# 5 Conclusions

The altitude distribution of BC in atmosphere plays a crucial role in deciding the BC-induced warming of the atmosphere. Such an atmospheric warming due to BC gets amplified when BC lies above strongly reflecting surface like clouds. The enhanced warming of BC can give rise to local instability and subsequent vertical lifting. Such absorption-warming-convection cycle can transport BC to higher altitudes. On account of its absorbing nature, high altitude BC can burn-off cirrus clouds. Realizing the importance of the vertical profile of BC, Babu et al. (2011) measured vertical profile of BC over strongly convecting Indian region during pre-monsoon periods (March 2010) using high altitude zero pressure balloon. The authors reported 2 sharp and confined high altitude layers of BC at 4.5 km and 8.3 km. The existence of the high altitude BC layers was confirmed by two subsequent balloon flights conducted during the subsequent winter and pre-monsoon seasons. While the high values of BC within first 3 km of atmosphere could be explained by boundary layer mixing, the higher BC values aloft appeared to be a mystery. The present study used the regional chemistry transport model - WRF-Chem, to understand the causes behind the observed high altitude sharp and confined BC layers. Firstly the simulations (incorporating surface BC emissions) of the meteorological parameters within the model were examined. Broad agreements of model simulated meteorological variables with their corresponding observations from the balloon measurements were found. Then the model simulated vertical profile of BC were compared with the corresponding observed profile, but it was found that the model could not replicate the high altitude BC layers as seen in observations. Thus reasonable simulations of meteorology within the model could not give rise to the high altitude BC layers. Then the emission of BC from high altitude sources i.e. aircrafts were prescribed in our simulations. The aircraft BC emissions from MACCity inventory, interpolated to our model grids with necessary modifications were used for this purpose. Upon the prescription of aircraft BC emissions from the regionally fine-tuned MACCity inventory, the model simulated vertical profile of BC started showing the mysterious high altitude BC peaks as seen in the observations. The sharp and confined nature of the simulated BC peaks showed large similarities with the observations, though their exact altitude and the magnitude differed vis-a-vis the observations. The fact that, aircraft emissions of BC being the primary cause behind the high altitude BC peaks, was re-confirmed when the peaks remained undisturbed, even after shutting down the surface level emissions of BC. The probability of occurrence of such high altitude (9-11 km) BC peak is found to be more than 40% over many locations within the model domain, during all the seasons except monsoon. During the monsoon season, the strong convection mixes BC throughout the atmospheric column, resulting into removal of such elevated sharp peaks. The corresponding atmospheric heating rates due to such observed and simulated profiles of BC were then computed. It was seen that such BC profiles could give rise to local instability and subsequent vertical mixing even at higher altitudes (~10-11 km). Such high altitude BC peaks would cause warming which could be amplified in the presence of highly reflecting clouds beneath them. This absorption-warming-convection cycle could possibly lift BC to even higher altitudes. In addition to such a 'self-lifting', it was seen in a separate model simulation done for the monsoonal month of July that, such high altitude BC were transported vertically up by the underlying strong convection over the Indian region. The transport was seen to be as high as up to 17.6 km i.e. the UTLS region. Many previous studies have reported that the near-tropopause conditions over the Indian region especially during monsoonal months are more favorable for the cross-tropopause transport of the tropospheric air-mass.

Thus this could act as a pathway for the lifted aircraft-emitted BC layers to reach the stratosphere. Many previous studies have also reported presence of BC in stratosphere. To understand this further, the vertical profiles of aerosol extinction coefficient from satellite measurements over the stratospheric heights across the Indian region were examined. The extinction coefficient values as high as $0.02 \, km^{-1}$, which were an order of magnitude greater than the average tropical stratospheric aerosol extinction coefficients were noticed. Though such layers were more visible during the monsoonal months, their presence during winters hinted at the possible co-operation between self-lifting and large-scale ascent over Indian region, in bringing the layers in the UTLS region. On the further examination, it was found that the particle depolarization ratio corresponding to the high values extinction coefficient were higher than 0.3. Such high values of PDR signify non-spherical nature of the involved species. Eliminating the possibility of mineral dust particles and ice crystals being responsible for such high values of PDR and finding from previous studies about the presence of non-spherical BC chain agglomerates in the UTLS region, it was concluded that such BC structures would possibly be responsible for the high values of PDR over the stratospheric altitudes. Once in stratosphere, BC can reside for a longer period of time due to the inherent stability of stratosphere and the absence of strong removal mechanisms. One of the possible implications of presence of BC in stratosphere is the reaction involving decomposition of ozone on BC, which would result in depletion of ozone. So potentially, the lifted upper tropospheric BC in stratosphere could harm the earth's protective blanket. Thus, while aircrafts look to be the cause for the sharp and confined high altitude layers of BC, such layers when lifted to stratosphere (under favorable conditions), can potentially affect the ozone layer and can have significant implications for health of all living organisms. Realizing the potential of emissions from aircrafts, a recent article (Editorial, 2016) also advocates for regulations on aircraft emissions. More observational studies using satellite and stratospheric balloons, and modeling studies are required to address this important phenomena especially over regions of high aircraft activity.

**Table 1.** Vertical levels prescribed in WRF-Chem simulations

| Level Index | Altitude (km) |
| --- | --- |
| 01-10 | 0-2 |
| 11-14 | 2-4 |
| 15-31 | 4-6 |
| 32-37 | 6-8 |
| 38-49 | 8-9 |
| 50-51 | 9-10 |
| 52-59 | 10-11 |
| 60-70 | 11-20 |

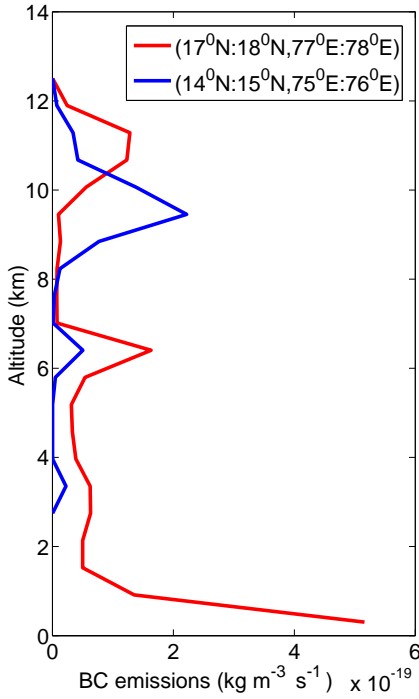

**Figure 1.** The vertical profile of BC emissions (kg m$^{-3}$ s$^{-1}$) from aircrafts in the MACCity inventory over $1^0 \times 1^0$ grid boxes a) centered over Hyderabad (red line) b) located $2^0$ south-west of Hyderabad (blue line) during the month of March, 2010.

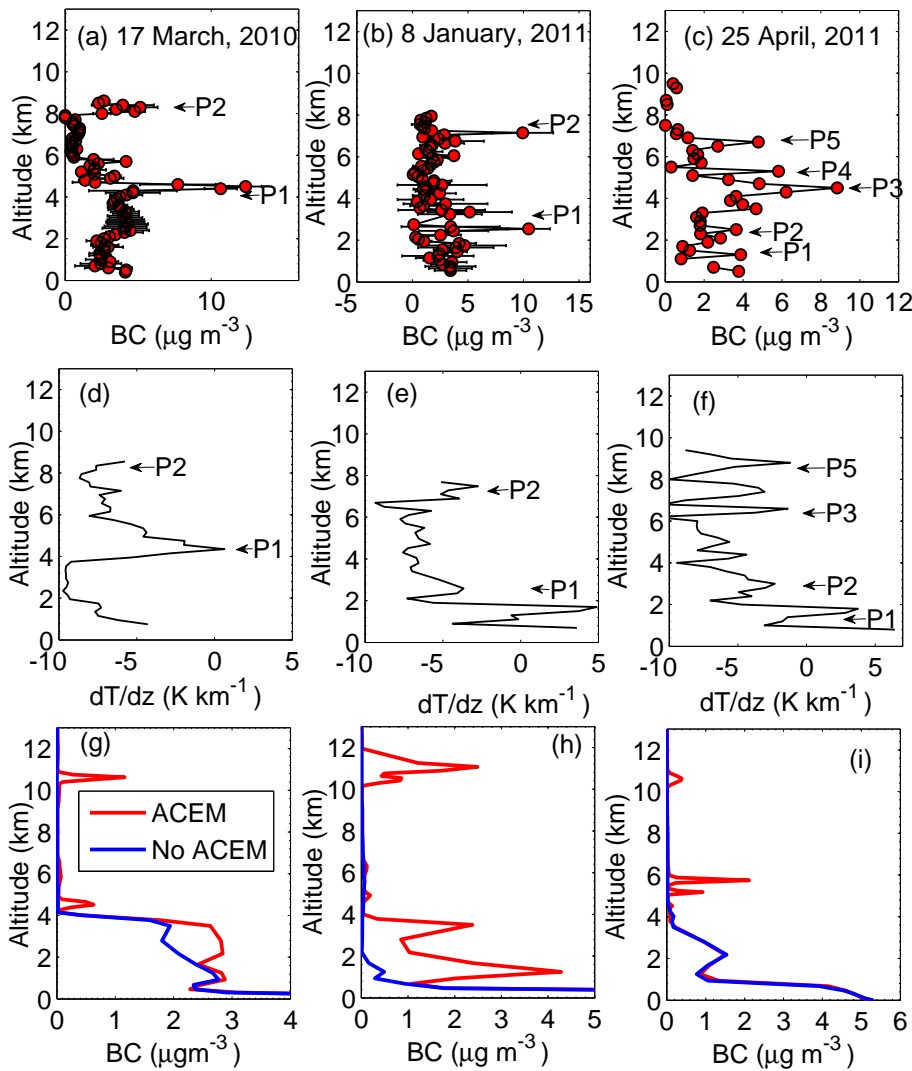

**Figure 2.** Observed vertical profile of BC over Hyderabad obtained from balloon measurements (a). during $17^{th}$ March 2010 (b). during $8^{th}$ January 2011 (c). during $25^{th}$ April 2011. The high altitude BC peaks in each profile are identified by letters P1 to P5. Observed vertical profile of dT/dz (K km$^{-1}$) over Hyderabad region obtained from balloon measurements (d). during $17^{th}$ March 2010 (e). during $8^{th}$ January 2011 (f). during $25^{th}$ April 2011. The locations corresponding to high altitude BC peaks are identified by letters P1 to P5. Model simulated vertical profile of BC over the area in the vicinity of the balloon flight region for (blue line) NoACEM/Ctrl (control run) and (red line) ACEM (runs with prescription of aircraft BC emissions) during (g). $17^{th}$ March 2010, (h). $8^{th}$ January 2011 and (i).$25^{th}$ April 2011.

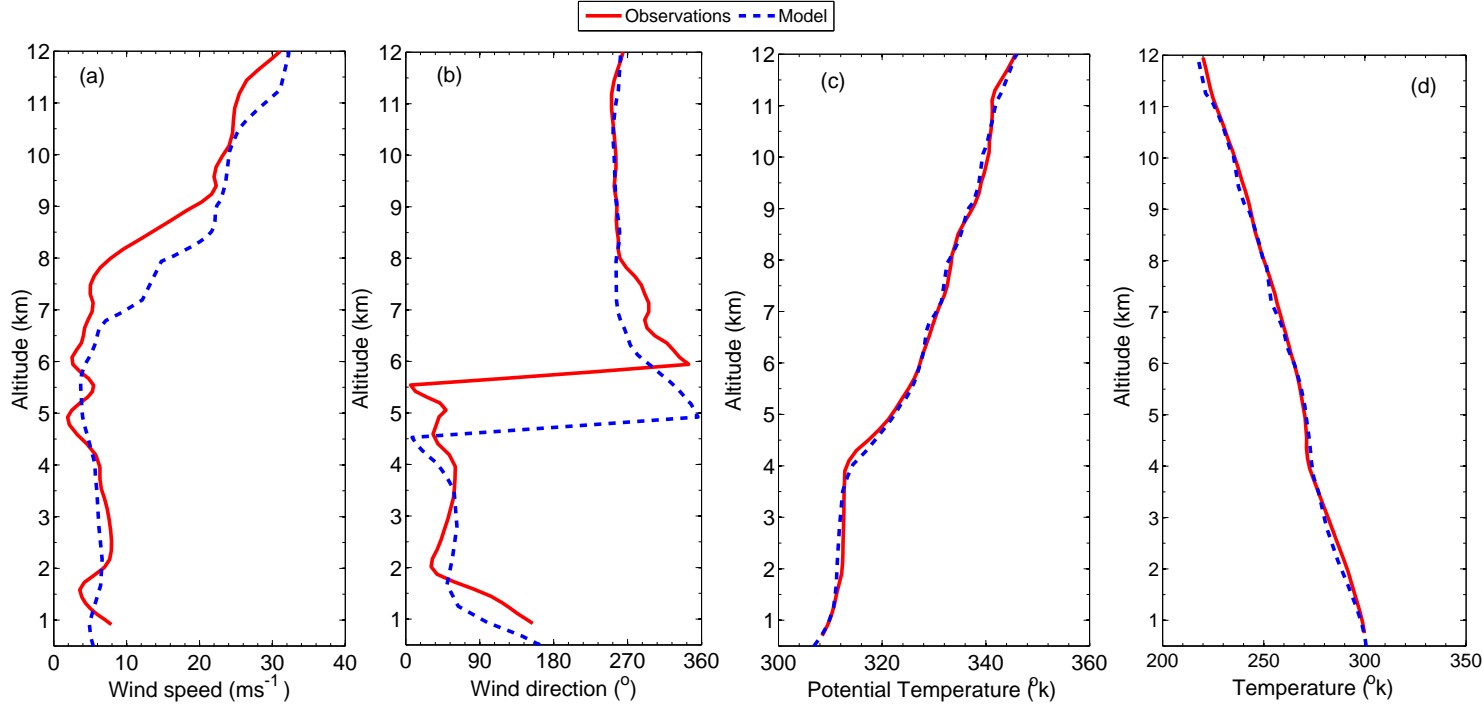

**Figure 3.** Comparison between the model simulated meteorological variables in NoACEM/Ctrl (control) run and the corresponding observations over the balloon flight region on $17^{th}$, March 2010 for (a) wind speed (b) wind direction (c) Potential temperature (d) Temperature

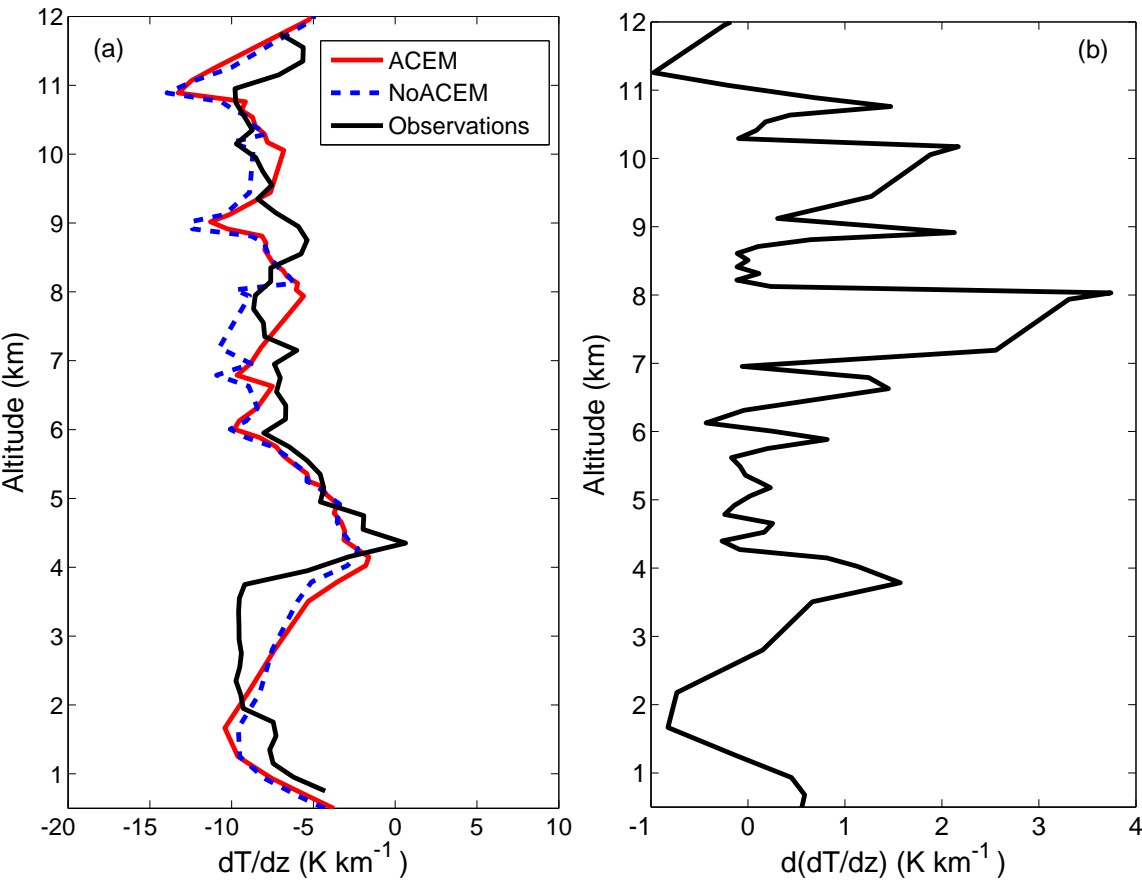

**Figure 4.** (a) Vertical profile of dT/dz (K km$^{-1}$) over the balloon flight region on $17^{th}$, March 2010 from: (black-line)- Observations, (blue dashed line)- Simulations without BC emissions from aircrafts (NoACEM) and (red line)- Simulations with BC emissions from aircrafts (ACEM). (b) Vertical profile of difference in dT/dz values for 2 model simulations, d(dT/dz)= dT/dz$_{ACEM}$ - (dT/dz)$_{NoACEM}$

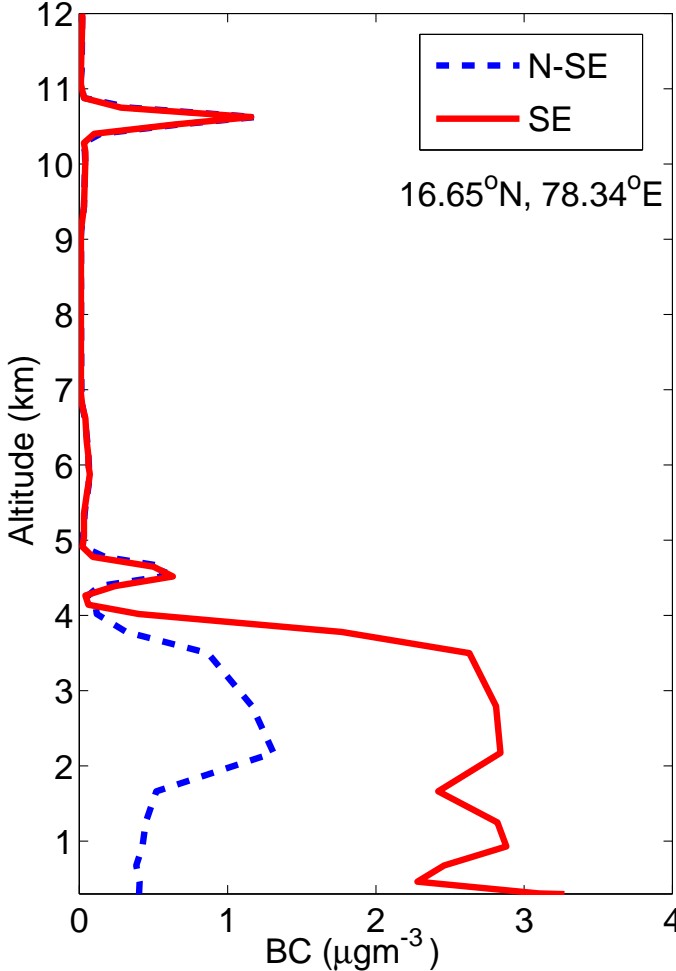

**Figure 5.** Model simulated vertical profile of BC during $17^{th}$, March 2010 for (a). N-SE runs (model simulations without prescription of surface level anthropogenic BC emissions) (b). SE runs (model simulations with prescription of surface level anthropogenic BC emissions).

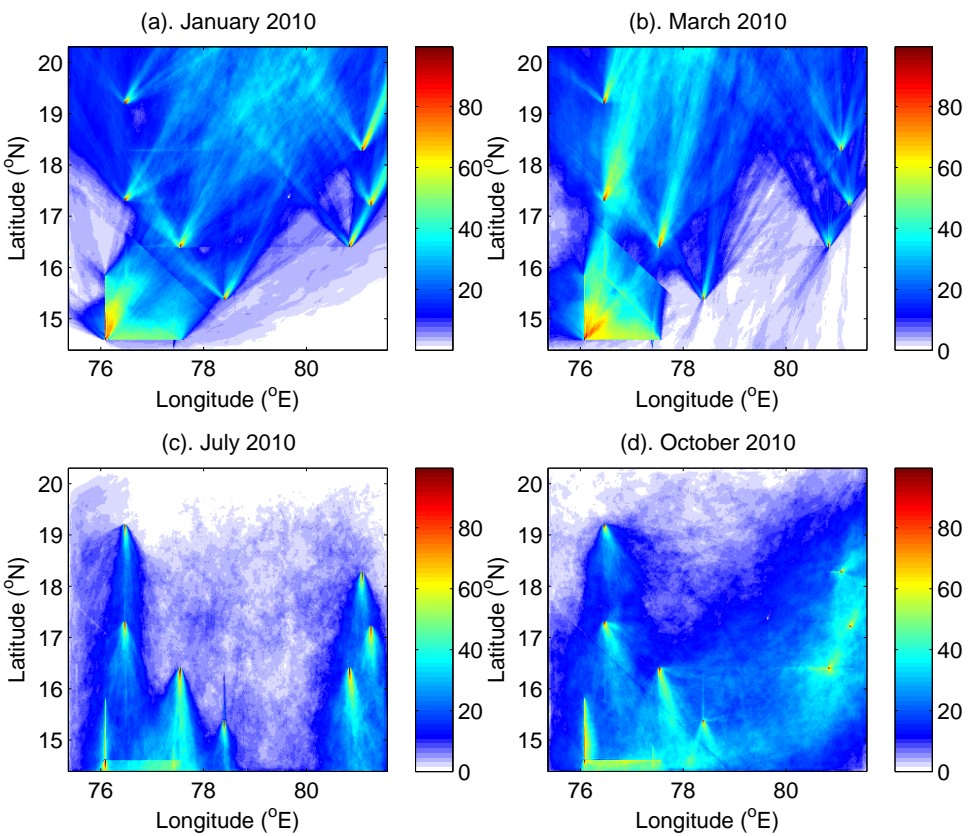

**Figure 6.** Spatial plot of probability (%) of capturing a peak in simulated BC within 9-11 km for the model simulations carried out during a). January 2010, b). March 2010, c). July, 2010 and d). October 2010

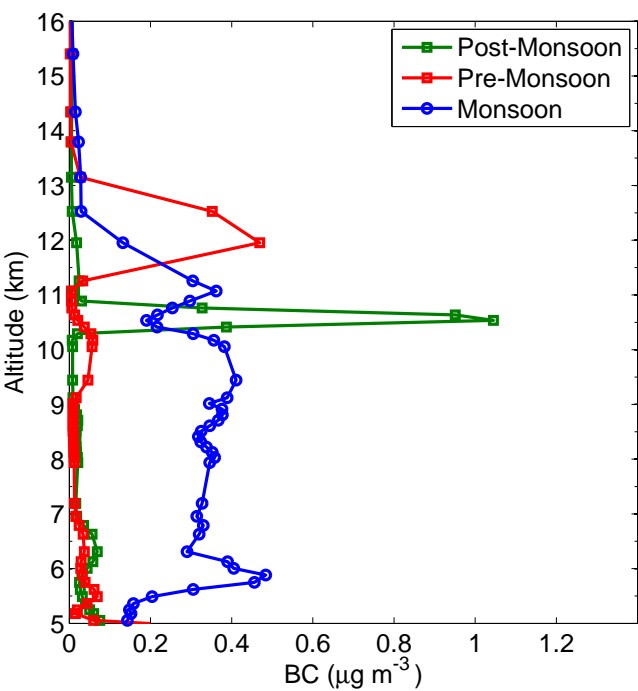

**Figure 7.** The model simulated vertical profile of BC mass concentration ($\mu$g m$^{-3}$) in the vicinity of balloon flight domain during different seasons, Pre-Monsoon (red line), Monsoon (blue line) and Post-Monsoon (green line), of the year 2010.

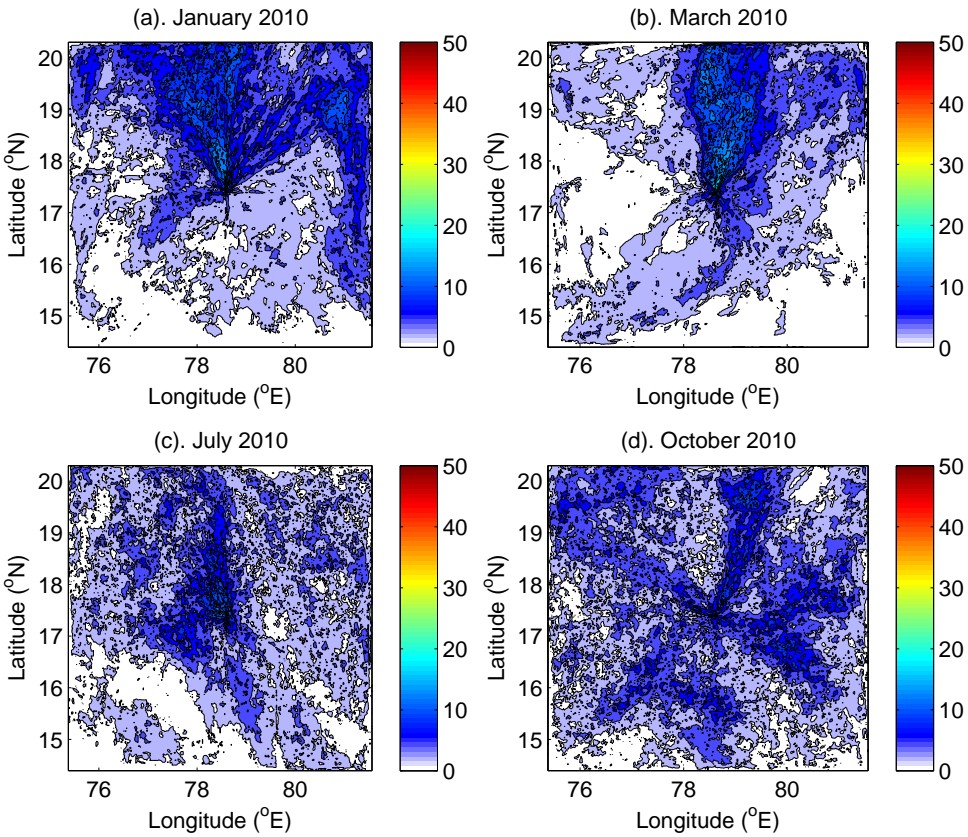

**Figure 8.** Spatial plot of probability (%) of capturing a peak in simulated BC within 4-5 km for the model simulations carried out during a). January 2010, b). March 2010, c). July, 2010 and d). October 2010

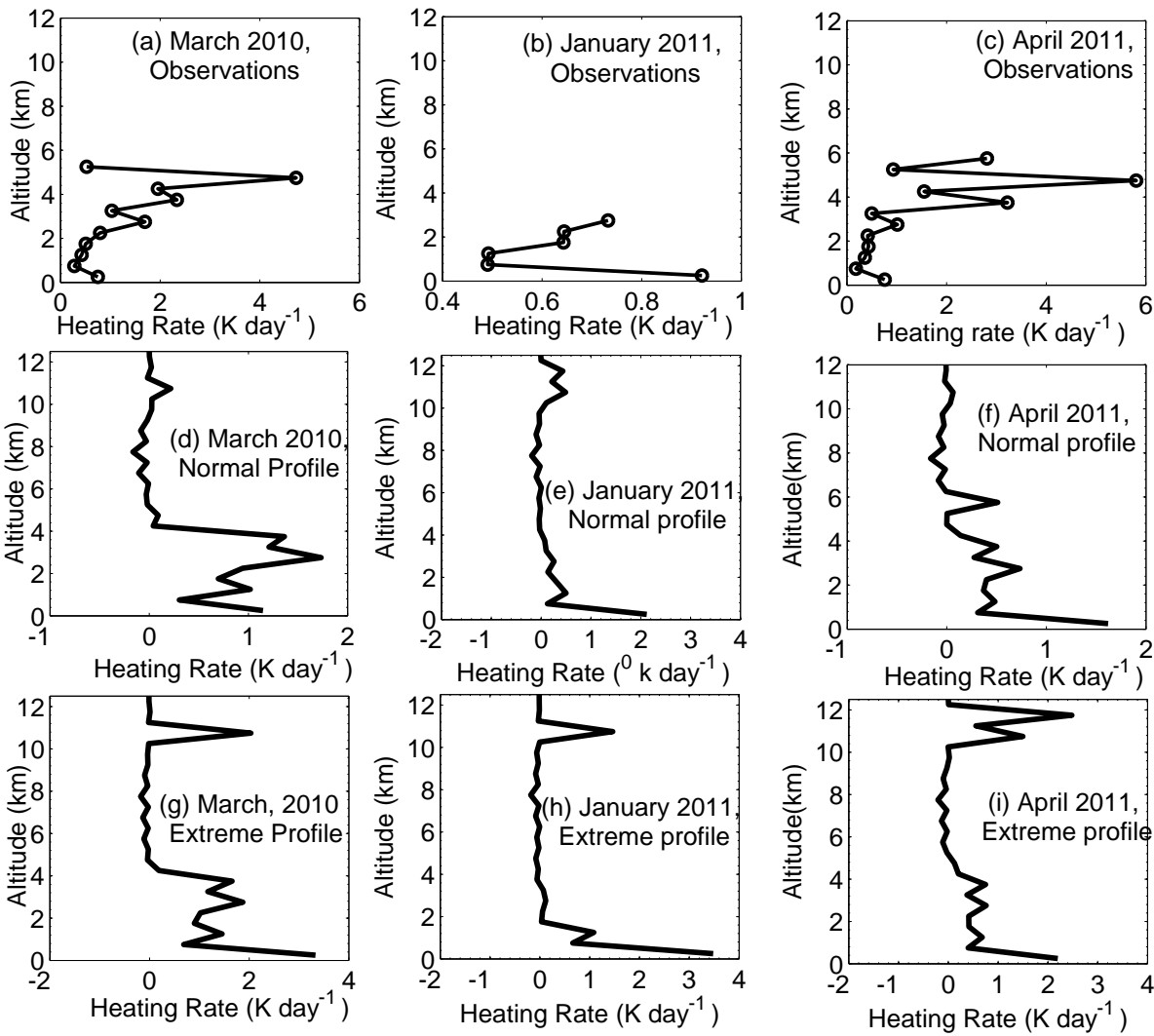

**Figure 9.** Vertical profiles of atmospheric heating rates for the observed BC profiles during (a). March 2010, (b). January 2011 and (c). April 2011. Vertical profiles of atmospheric heating rates corresponding to the observed BC profiles during (a). March 2010, (b). January 2011 and (c). April 2011. Vertical profiles of atmospheric heating rates corresponding to the model simulated BC profiles during (d and g). $17^{th}$, March 2010 (e and h). $8^{th}$, January 2011 (f and i). $25^{th}$, April 2011.

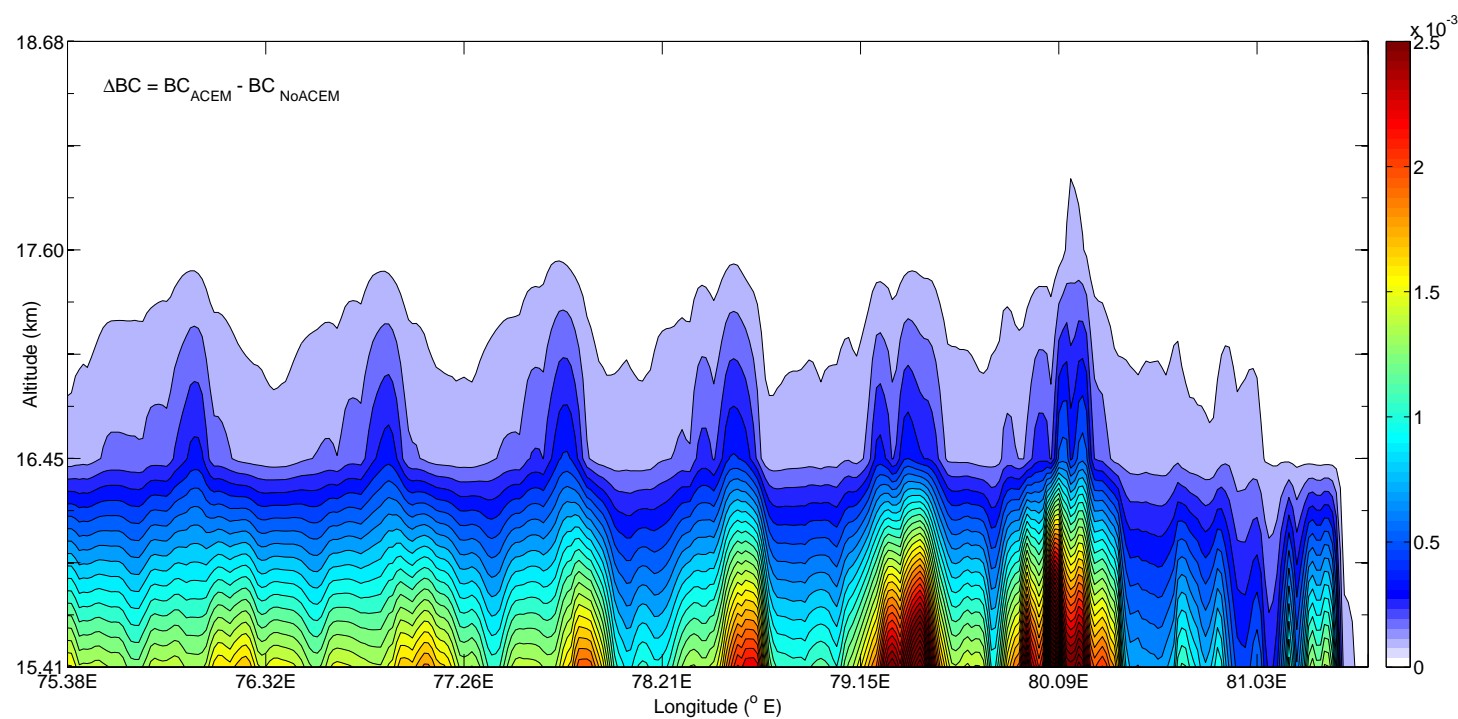

**Figure 10.** Vertical profile of the difference in BC mass concentration ($\mu$g m$^{-3}$) in ACEM runs vis-a-vis NoACEM run, averaged for the entire duration of the model simulation, carried out during July 2010. Here, $\Delta$BC= BC$_{ACEM}$-BC$_{NoACEM}$.

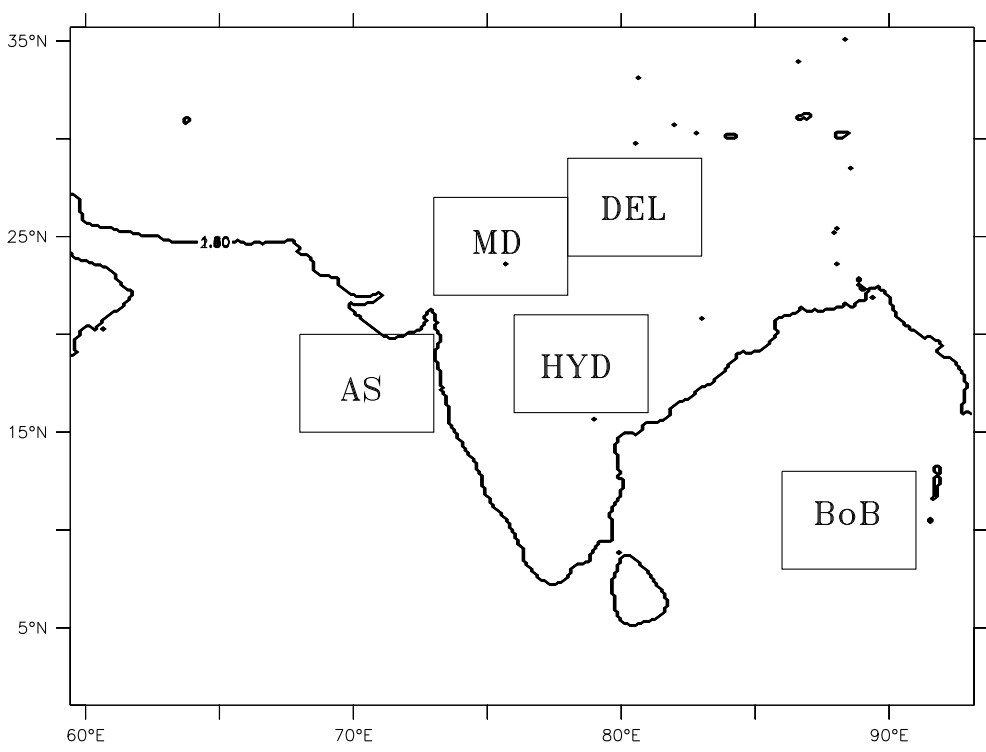

**Figure 11.** $5^0 \times 5^0$ regional boxes considered for the CALIPSO analysis (AS-Arabian sea, BoB-Bay of Bengal, HYD- Hyderabad, MD-Mumbai-Delhi and DEL-Delhi).

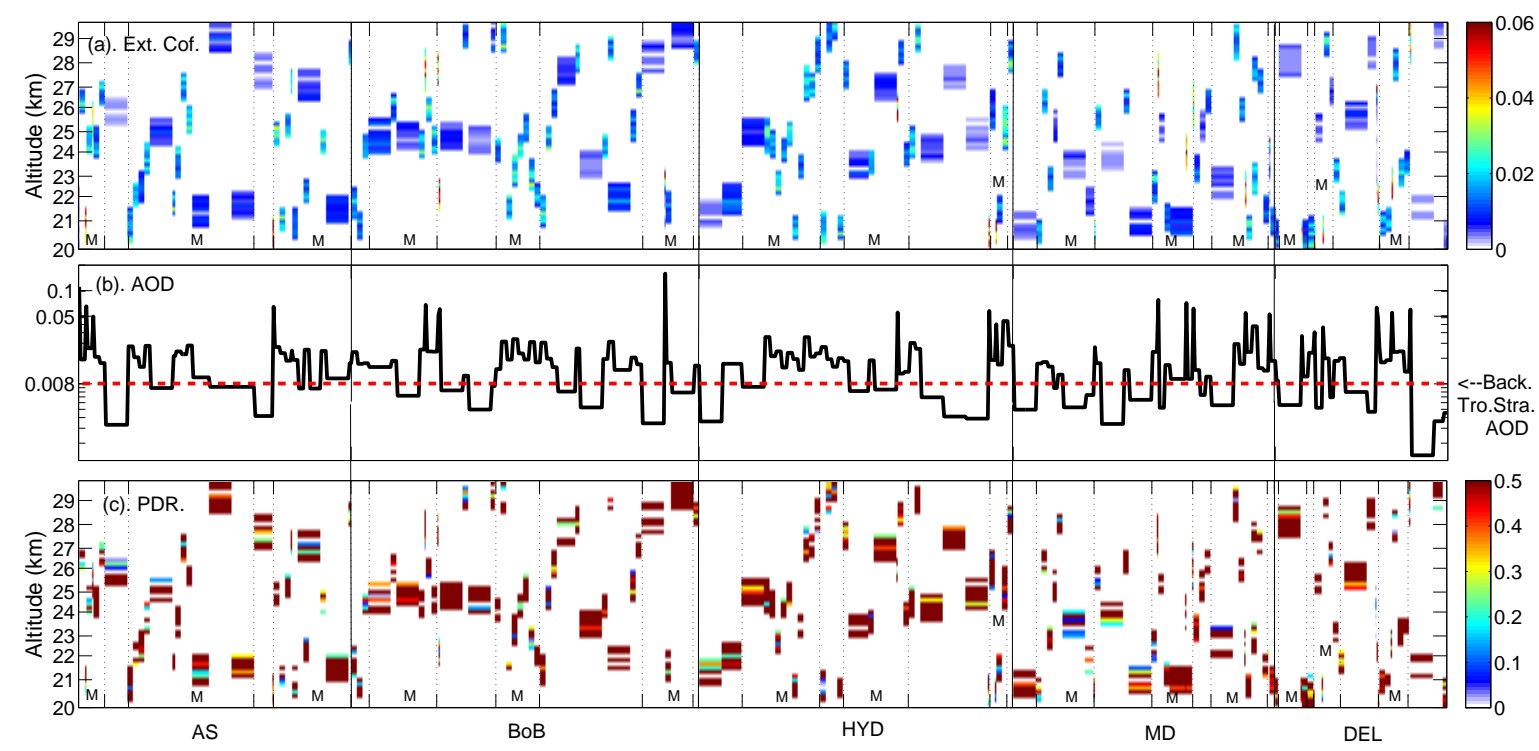

**Figure 12.** Time series of vertical profile of (a). Extinction coefficient over 5 regions (AS-Arabian sea, BoB- Bay of Bengal, HYD- Hyderabad, MD- Mumbai-Delhi and DEL-Delhi) over and around India (b). Corresponding stratospheric (from 20 km to ∼30 km) AOD, the background stratospheric AOD for the entire tropical belt (Kremser et al., 2016) is shown by the dotted red line (c). Particle Depolarization Ratio (PDR) over the same 5 regions from January 2010 to December 2012. The individual time series of Ext. Cof., AOD and PDR. of each region are appended together in the figure one after the other. The letter M signifies the Monsoon (JJAS) season during a year under consideration

## 6 Code availability

The simulations are done using an open source online regional chemistry transport model WRF-Chem, which is freely available at http://www2.mmm.ucar.edu/wrf/users/.

## 7 Data availability

5 The extinction coefficient level 2 data from CALIPSO is available freely on https://eosweb.larc.nasa.gov/HBDOCS/langley_web_tool.html.

*Competing interests.* The authors declare that they have no conflict of interest

*Acknowledgements.* All the computations are conducted on the computational cluster funded jointly by Department of Science and Technologies FIST program (DST-FIST), Divecha Center for Climate Change and ARFI project of Indian Space Research Organisation (ISRO).
10 This work is partially supported by MoES (Grant No:MM/NERC-MoES-1/2014/002) under the South West Asian Aerosol Monsoon Interactions (SWAAMI) project. We thank the anonymous reviewers for the detailed evaluation and useful suggestions. We would also like to thank the Computational and Information Systems Laboratory (CISL-NCAR) for the Research Data Archive.

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
