# Peer review of "Possible climatic implications of high altitude emissions of black carbon"

_Atmospheric Chemistry and Physics, 2017_

## Referee Comment (RC1) · Anonymous Referee #1 · 13 Mar 2017

General comments:

The authors first showed vertical distributions of BC simulated by WRF-Chem around Hyderabad/India for three different days (17 March 2010, 08 January 2011, and 25 April 2011). The previous study showed that the observed BC has prominent spikes in 4 km or higher altitudes. The authors identified that the source of the spikes of BC is the aircraft emission. Second, the authors confirmed that the BC particles from the aircraft emission reach in lower stratosphere by using both the simulation and observation of CALIPSO. The manuscript and logic are straightforward, but I would like to ask the authors to add more explanation for more deeply understanding. Especially, how much are the results representative in each season? Although the measurements are very limited, i.e., one day per one season, the simulation seems to be easily conducted in at least one month to generalize and strengthen the conclusions. In overall, the

manuscript would be acceptable for publication if these comments can be satisfactorily addressed.

Specific comments:

1. P.2, L27: Could the authors clarify actual heights of "the middle troposphere" here?

2. P.5, L2: How do the authors classify the BC1 (hydrophobic) and BC2 (hydrophilic)? Do the authors consider atmospheric aging processes of BC?

3. P.7, L32-33: The modification is not clear. Especially, the readers cannot understand the vertical profile of the BC emission. Please clarify it.

4. P.13, L20-22: How do the authors determine the background tropical stratospheric AOD? Please add some references or evidence to the manuscript.

5. P.14, L10-11: The reason to eliminate the existence of dust is not unclear for me, because there is a possibility to existence of dust particles in the fine mode. The fine particles of dust perhaps exist in the 21-22 km layers. Could the authors add more evidence or include the possibility of mineral dust to strengthen the reason of the elimination of dust in your analysis?

6. P.15, L10: Do the authors have any evidence of the increasing aircraft traffic?

7. Fig 4: Is this weather a typical around Hyderabad. Also, how about the other days (8 January 2011 and 25 April 2011)?

8. Figs 5 and 6: What is the difference in NoACEM/Ctrl between Fig 5 and Fig 6? I am confused.

9. Fig 7: How much did the meteorological fields, i.e., air temperature, change by implementing the BC radiative impacts?

Technical corrections:

1. P.4, L23: How do the authors define the 70-vertical layer? Please clarify each level.

2. P.8, L21: The acronym "FINN" should be defined in L20 not L21.

3. Fig 3: The map is too simple to know the information about city name and topography. Most of readers are not familiar to this area.

---

## Referee Comment (RC2) · Anonymous Referee #2 · 13 Mar 2017

This study investigates the reasons behind measured high-altitude BC layers over India for 3 seperate days using a chemistry transport model. The topic is relevant and within the scope of ACP. The introduction is clear and well-written. When it comes to the general presentation of the results and the figures, however, I think this manuscript needs further work before I can recommend it for publication. First, the aircraft emissions in this study were scaled, but this is only mentioned in one sentence and never mentioned again in the abstract or conclusion. The authors correctly states that the uncertainties in emissions are large, but it is not clear to me why the emissions were scaled, by how much, and how this could be justified. Second, the manuscript is written like a story ('then we did this and then we did that'), and it needs to be rewritten to a more common form with 'Methods' and Results' clearly separated. I also miss a 'Discussion' section where the authors discuss some of the uncertainties with their study, e.g. the limitation

of data (only 3 days), the scaling of the emissions etc. Third, the figures need to be plotted on the same format, and some of them can be combined or removed. Finally, I think there is a tendency for the interpretation of the results to be slightly overstated. The study itself is interesting, and it would read much better with a plain description of the results, without any 'convincing'. There are also use of words like 'significance difference' without any significance testing mentioned.

Specific comments:

Abstract: Please specify the 3 dates for the measurements, and that you scaled the aircraft emissions.

Fig1: What does the error bars illustrate? Do you know why the error bars are much larger in a) and b) compared to c)?

Chapter 2: Can you add a small paragraph on how BC were measured using an Aethalometer, e.g. filter-based absorption. What are the uncertainties on separating out other absorbing components, like dust? Can you also report on the MAC that was used? I suggest specifying that 'elemental carbon' is measured (eBC), as suggested in literature (Petzold et al 2013 and references within).

I suggest to drop Fig3.

There are many figures and I think it would be easier to read if some of the figures could be combined. For instance, maybe Fig1 and Fig2 and Fig5/6 can be merged into one bigger figure (with 9 equally sized panels) so it is easier to compare the peaks? In that case; can you plot the panels on the same y axis, some goes up to 10 km vs. 9 km vs 12km. I find it confusing that the axis and panels are plotted differently for each figure, e.g. comparing Fig 5 and Fig6, where the black line in Fig5 are the same as blue line in Fig6 (?). I suggest dropping Fig5 as the same info is in Fig6. Fig 9 and Fig 10 can also be combined (with 9 panels).

P6 L16: Can you report numbers for the differences in magnitudes you are referring

to? It is hard to read on the figure.

P6 L25: What do you mean by 'the simulated BC significantly match the observation'? Did you do any significance testing on this? Again, it is difficult to compare the model and the observations since the plots are so different (axis).

P7 L5: What do you mean by the meteorology being benign? And that the long-range transport of BC is unlikely?

Your manuscript is written more like a story, and how you came about with the different hypothesis, but I suggest rewriting the manuscript with the more common separation of 'Methods' and 'Results', which means e.g. that 4.3 and 4.6 (until L15 on P10) should be moved to 'Methods' along with everything else in Results that explains the methods you have used. Parts of 4.7 explaining background for convective lifting can be moved to introduction (and the rest to methods). Same goes for Conclusions.

P7 L32: This is the first time and only place you say that you have scaled the aircraft emissions you have used. How did you modify the emissions? You also need to mention this in the abstract/introduction, throughout the text and in the conclusions.

P9 L29: How is 'soot' defined in this context?

P8 L7: 'the nature of the layers (sharp and confined) looks very similar'. Please rewrite. Also, the peaks are much smaller in magnitude and are not located at the same altitudes as the observations?

P8 L11: 'This clearly highlights the role played by aircraft emissions' However, you did scale the emissions? You need to emphasis this more. Do you see the peaks when you run with unscaled emissions?

P8 L34: 'Beyond 4 km, the profiles are identical'. It is hard to say this without any significance testing.

P9 L23: '..the observed BC profiles causes more heating near the surface'. Did you

show that the observed BC profiles cause this heating?

P10 L29: 'The identical nature of (..), brings out the average pattern'. I do not understand this sentence.

P11 L21: I am not sure if I understand your explanation of 'normal' and 'extreme' profiles in terms of the starting point. Area in the model?

Fig 13 is very difficult to read.

Technical corrections:

P2 L1: What is 'such an' refer to here? I suggest to remove.

P2 L12: Can you provide references to precip/convection?

P6 L3: 1st – first

P6 L8: either remove 'etc..' or replace by explaining the other discrepancies.

---

## Author Comment (AC1) · 22 May 2017

**Possible climatic implications of high altitude emissions of black carbon**

Gaurav Govardhan[1], Sreedharan Krishnakumari Satheesh[1,2], Ravi Nanjundiah[1,2], Krishnaswamy Krishna Moorthy[1], and Surendran Suresh Babu[3]

[1]Center for Atmospheric and Oceanic Sciences, Indian Institute of Science, Bengaluru, India
[2]Divecha Center for Climate Change, Indian Institute of Science, Bengaluru, India
[3]Space Physics Laboratory, Vikram Sarabhai Space Centre, Thiruvananthapuram, India

*Correspondence to:* Gaurav R. Govardhan (govardhan.gaurav@gmail.com)

**Authors' reply to comments from the first reviewer**

First of all, we would like to thank the anonymous reviewer for appreciating our work, giving constructive suggestions and overall positive recommendations.

**General comments from the reviewer:**

The authors first showed vertical distributions of BC simulated by WRF-Chem around Hyderabad/India for three different days (17 March 2010, 08 January 2011, and 25 April 2011). The previous study showed that the observed BC has prominent spikes in 4 km or higher altitudes. The authors identified that the source of the spikes of BC is the aircraft emission. Second,

10  the authors confirmed that the BC particles from the aircraft emission reach in lower stratosphere by using both the simulation and observation of CALIPSO. The manuscript and logic are straightforward, but I would like to ask the authors to add more explanation for more deeply understanding.

Especially, how much are the results representative in each season? Although the measurements are very limited, i.e., one

15  day per one season, the simulation seems to be easily conducted in at least one month to generalize and strengthen the conclusions. In overall, the manuscript would be acceptable for publication if these comments can be satisfactorily addressed.

**Authors' reply:**

We appreciate this suggestion from the reviewer regarding seasonal simulations. As suggested by the reviewer, we have car-

20  ried out 1-month long model simulations for each season i.e. winter, pre-monsoon, monsoon and post-monsoon, of the year 2010. The months January, March, July and October are taken as representatives of the respective seasons. The analysis of the monthly mean vertical profiles of BC from the model simulations brings out following main results:

1. During the winter, pre-monsoon and the post-monsoon seasons, a large number of locations within the model domain show more than 50% probability of occurrence of a high altitude (9-11 km) BC peak (a few locations show more than 60% proba-

25  bility).

2. The upper levels winds and the aircraft BC emissions themselves play a major role in deciding the horizontal location of the high altitude (9-11 km) BC peaks.

3. During the monsoon season, such probability values and the locations with high-altitude (9-11 km) BC peak reduce. The main reason for such reduced occurrence of high altitude BC peaks during the monsoon season appears to be the monsoonal convective lifting of BC which uniformly distributes BC at the upper levels (blue line, fig.1), which increases the total BC mass concentration at the upper levels, while getting rid of the sharp BC peaks.

[Figure]

**Figure 1.** The model simulated vertical profile of BC mass concentration ($\mu$g m$^{-3}$) in the vicinity of balloon flight domain during different seasons, Pre-Monsoon (red line), Monsoon (blue line) and Post-Monsoon (green line), of the year 2010.

The detailed analysis of the seasonal simulations will be included in the modified manuscript.

**Specific comments from the reviewer:**

1. **P.2, L27:**
   **- Original sentence from the manuscript:**
   Synthesizing multi-platform measurements Satheesh et al. (2008) have revealed the existence of elevated aerosol layers in the middle troposphere.
   - Reviewer's comment:
   Could the authors clarify actual heights of "the middle troposphere" here?

Here "the middle troposphere" refers to a height of 4 to 5 km.

2. **P.5, L2**

   **- Original sentence from the manuscript:**

   The model takes into account the following aerosols species: BC1 (Hydrophobic), BC2 (Hydrophilic)

   -Reviewer's comment:

   How do the authors classify the BC1 (hydrophobic) and BC2 (hydrophilic)? Do the authors consider atmospheric aging processes of BC?

   - Authors' reply:

   The regional chemistry transport model WRF-Chem, used for these simulations considers black carbon aerosol to be present in two different modes: hydrophobic (BC1) and hydrophilic (BC2). The characteristic conversion e-folding lifetime from hydrophobic to hydrophilic i.e. BC1 to BC2 is considered to be 2.5 days. The primary emissions of BC are assumed to occur in the hydrophobic mode (BC1). The BC is assumed to be removed from atmosphere by dry deposition (for both the hydrophobic and hydrophilic modes) and wet deposition (for the hydrophilic mode). More details about the treatment of BC in WRF-Chem can be found Kumar et al. (2015).

3. **P.7, L32-33:**

   **- Original sentence from the manuscript:**

   Thus, considering such underestimations of EI(BC) in the current emissions inventory, we modified the emissions and forced our model with such emissions at 23 levels with appropriate mapping to model levels.

   -Reviewer's comment:

   The modification is not clear. Especially, the readers cannot understand the vertical profile of the BC emission. Please clarify it.

   -Authors' reply:

   **a). Vertical profile of the BC emissions from the aircrafts within the MACCity inventory:**

   The vertical profile of BC emissions from aircrafts, averaged over a $1^0 \times 1^0$ grid box centering Hyderabad (fig.2, red line) and that over another grid box (fig.2, blue line) of the same dimension just $2^0$ south-west of the Hyderabad grid box, are shown in fig.2. It could be clearly noted that the vertical profile of BC emissions from aircrafts shows multiple-peaks over the Hyderabad region (fig.2, red line), with highest emissions near the surface followed by a peak at 6 km and at 11 km. On the other hand, over the other grid box which is relatively farther from the airport location, the vertical profile of BC emissions from aircrafts (fig.2, blue line) shows more dominant peak at higher levels (10 km) with relatively lesser BC emissions at 6 km and 4 km altitudes. The near-surface peak in BC over Hyderabad (fig.2, red line) could be due to the Landing and Take-Off (LTO) activities within the airport. The peaks at an altitude of 10 km (fig.2, red line) could be due to the aircrafts which are flying at the cruising altitudes and not landing at or taking-off from Hyderabad airport.

[Figure]

**Figure 2.** The vertical profile of BC emissions (kg m$^{-3}$ s$^{-1}$) from aircrafts in the MACCity inventory over $1^0 \times 1^0$ grid boxes a) centered over Hyderabad (red line) b) located $2^0$ south-west of Hyderabad (blue line) during the month of March, 2010.

**b). Modification of aircraft BC emissions:**

The MACCity emission inventory database (Lamarque et al., 2010), used in our study for the emissions of BC from aircrafts, is prepared using AERO2k database of global aircraft movement (Eyers et al., 2004), fuel consumption from PIANO aircraft emission tool (Simos, 2004) and the BC emission factors reported by Eyers et al. (2004). To assess this inventory over Hyderabad (our study region), we estimated BC emissions from aircrafts over this region using previously reported values for fuel efficiency of different airline carriers (Kwan and Rutherford, 2015), seating capacity of the carriers (websites of different airlines), density of aviation fuel used (Cookson and Smith, 1990; Arkoudeas et al., 2003; Outcalt et al., 2009; Blakey et al., 2011) in Indian region, BC emission index for the aviation fuel (Stettler et al., 2013), and actual data on air traffic over Hyderabad obtained from Air Traffic Controller, Hyderabad (Babu et al., 2011), following the method given below.

In view of the reliable information of the fuel efficiency of different airline carriers in India, we used the fuel efficiency of Trans-Atlantic aircrafts, reported by Kwan and Rutherford (2015), which gives values in the range 27 passenger-km/L for British Airways to 40 passenger-km/L for Norwegian airlines (Passenger – km per litre = No. of passengers in an aircraft × km/L for 1 passenger). We also estimated the average seating capacity of different aircrafts (regional, shorthaul, medium haul, long-haul etc) of the various airlines in India from their respective websites, and these are given in Table.1.

**Table 1.** Calculations regarding BC emitted by aircrafts in our single grid volume.

| Type of air journey (haul) | Average number of seats | Average number of passengers | Fuel consumed per km by an aircraft (L/km) | Fuel consumed per km by an aircraft (kg/km) | BC emitted ($\mu$g m$^{-3}$) by an aircraft in our model grid $\times 10^{-3}$ | BC emitted ($\mu$g m$^{-3}$ s$^{-1}$) by aircrafts in our single grid volume $\times 10^{-6}$ |
|---|---|---|---|---|---|---|
| Regional | 186 | 140 | 4.376 | 3.413 | 1.5 | 3.338 |
| Short | 181 | 135 | 4.234 | 3.3 | 1.452 | 3.22 |
| Medium | 288 | 216 | 6.774 | 5.28 | 2.325 | 5.167 |
| Long | 357 | 268 | 8.39 | 6.545 | 2.88 | 6.398 |

We used an average load factor of 75% for obtaining the values in the third column, following Bhullar (2017), which reports the average load factor for the year 2011-12 for the Indian domestic flights. Using this average passenger load, we estimated the fuel consumed in L/km is each case and these are given in the fourth column of the table 1. The mean density of aircraft fuel is taken as 780 kg m$^{-3}$ (Cookson and Smith, 1990; Arkoudeas et al., 2003; Outcalt et al., 2009; Blakey et al., 2011) and the fuel consumed in kg/km for the different aircrafts are calculated and given in fifth column of table 1. These values are further used to estimate the quantity of BC emitted. However the BC emission index, (the amount of BC emitted per kg of aviation fuel) has a large uncertainty and several studies (Herndon et al., 2008; Onasch et al., 2009; Timko et al., 2010) have shown that, this parameter spans 4 orders of magnitude. Stettler et al. (2011) suggests that the currently used emissions inventories could underestimate EI(BC) by an order of magnitude or more. In this study, we have used an EI(BC) value of 0.088 g(BC)/kg of aircraft fuel burnt as suggested in a recent study by Stettler et al. (2013). Using the aforementioned EI, the emission intensity within a single grid volume of our model (i.e. 2000m $\times$ 2000m $\times$ 100m ), if an aircraft crosses the grid volume, is estimated for different air journey types and is given in the seventh column of table1. Using the above information, and the information obtained on the number of aircrafts that used the Hyderabad flight corridor (as per the information obtained from the ATC reported in Babu et al. (2011)), we made an estimate of the average BC emission rate for each aircraft type as given in column 7, table 1. Using the mean of seventh column, table 1 and assuming 100 km$^2$ as the influence zone of Hyderabad ATC, we estimate the total emissions of BC over the Hyderabad region within the flight corridor of 8-10 km to be 2.2536 $\times$ 10$^{-4}$ $\mu$g m$^{-3}$ s$^{-1}$ . The corresponding total emissions within the flight corridor over Hyderabad region from MACCity inventory comes out to be 3.606 $\times$ 10$^{-7}$ $\mu$g m$^{-3}$ s$^{-1}$, which is far lesser than our estimates. Thus one scaling factor comes from this comparison. Additionally, acknowledging the coarse resolution of the MACCity inventory, the 'line-source' nature of the freshly emitted aircraft trail and the finer resolution of our model simulations, we confine the aircraft BC emissions into a width of 2 km and a height of 100 m. The mass conservation of the emitted BC due to such confinement leads

to an additional scaling factor. The total scaling factor becomes the product of these two scaling factors. The modified emissions are formulated by multiplying the original emissions by the aforementioned scaling factors.

4. **P.13, L20-22:**

   - **Original sentence from the manuscript:**

   It can be seen that, the stratospheric AODs from our analysis over the 5 regional boxes are roughly 4-6 times higher than the background tropical stratospheric AOD.

   -Reviewer's comment:

   How do the authors determine the background tropical stratospheric AOD? Please add some references or evidence to the manuscript.

   -Authors' reply:

   In our study, the background tropical stratospheric aerosol conditions are taken from a recent paper by Kremser et al. (2016), which reviews the current knowledge regarding stratospheric aerosols. The paper reports mean stratospheric (tropopause to 40 km) AOD over the tropics ($20^0$S-$20^0$N) for the years 1985 to 2012 (Figure 4 (bottom), Kremser et al. (2016)). The corresponding values are taken from SAGE II (The Stratospheric Aerosol and Gas Experiment II) instrument for the years 1985-2005, GOMOS (Global Ozone Monitoring by Occultation of Stars) satellite for 2002 to 2010, and CALIPSO (Cloud-Aerosol Lidar and Infrared Pathfinder Satellite Observations) satellite for 2006-2012. In our study, we examine the mean tropical stratospheric AOD from Kremser et al. (2016) for the years 2008 to 2012. The maximum value of the mean tropical stratospheric AOD during that period is considered to be the background tropical stratospheric AOD in our study.

5. **P.14, L10-11:**

   -**Original sentence from the manuscript:**

   One of the major non-spherical aerosol species over this region is mineral dust. But dust is relatively too heavy and coarse to be lifted up to such altitudes.

   -Reviewer's comment:

   The reason to eliminate the existence of dust is not unclear for me, because there is a possibility to existence of dust particles in the fine mode. The fine particles of dust perhaps exist in the 21-22 km layers. Could the authors add more evidence or include the possibility of mineral dust to strengthen the reason of the elimination of dust in your analysis?

   -Authors' reply:

   The source for mineral dust aerosol species in stratosphere could be largely related to a) volcanic eruptions b) meteoritic debris and c) convective transport of tropospheric dust. As mentioned in the manuscript our study has been carried out for a period which is relatively volcanically quiescent. Thus, volcanically erupted dust may not have contributed much to the stratospheric aerosol burden during our study period. The meteoritic debris form a minor part (5-10%) of the stratospheric aerosol composition, especially at altitude less than 30 km (Turco et al., 1981), making them difficult to be captured by a LIDAR. Moreover, the meteoritic debris are reported to be coarse in size (i.e. having radius above 1 $\mu$m)

h

**Table 2.** Year to year change in global air traffic reported by Airports Council International (ACI).

| Year | % change in passenger traffic | % change in air cargo traffic | % change in flight movements |
|------|-------------------------------|-------------------------------|------------------------------|
| 2013 | 4.6 | 0.9 | 0.6 |
| 2014 | 5.7 | 4.7 | 1.3 |
| 2015 | 6.4 | 2.6 | 2 |

(Turco et al., 1981; Mackinnon et al., 1982) (and hence would have large deposition rates) and are largely spherical in shape (Mackinnon et al., 1982) with lower values of particle depolarisation ratio (PDR less than 0.1 (Klekociuk et al., 2005)). The extinction coefficient associated with plume of meteoritic debris are 3 orders of magnitude lesser (Gorkavyi et al., 2013) than the values we notice over our study domain. These points together rule out the possibility of associating the observed values of extinction coefficient and PDR to the meteoritic dust. The convectively lifted dust and other air pollutants while can get into the higher altitude regime (Andreae et al., 2001; Randel et al., 2010; Vernier et al., 2011; Fadnavis et al., 2013; Corr et al., 2016), their transport over the Indian region is limited to around 20 km (Fadnavis et al., 2013), which is mainly governed by the height of convective towers (Meenu et al., 2010). Moreover, dust aerosol is 2.6 times heavier than BC (Hess et al., 1998) and also less solar absorbent; thus it is unlikely to get vertically lifted up beyond 20 km by large-scale or self lifting mechanisms (de Laat et al., 2012). Hence we eliminate the possibility of existence of tropospheric dust as well beyond 22 km altitude over our domain.

6. **P.15, L10:**

**Original sentence from the manuscript:**

The continued negative trend in stratospheric ozone over $40^0$S-$40^0$N from 1984 to 2014 (Bourassa et al., 2014) could possibly be related to the ever increasing aircraft traffic.

-Reviewer's comment:

Do the authors have any evidence of the increasing aircraft traffic?

-Authors' reply:

Airports Council International (ACI) (an international organization which is a representative of the world's airport authorities) reports following increasing trends (table 2) in global passenger traffic, cargo traffic and aircraft movements for the years 2013, 2014 and 2015 in their annual reports (Paraschis and Gittens, 2014; Piccolo and Gittens, 2015, 2016). Moreover, a continuous increasing trend in global passenger traffic has also been reported from the year 2003 (Paraschis and Gittens, 2014). These together confirm an ever increasing trend in global air-traffic for more than a decade.

7. **Fig 4:**

   **Figure from the manuscript:**

[Figure]

**Figure 3.** Comparison between the model simulated meteorological variables in NoACEM/Ctrl (control) run and the corresponding observations over the balloon flight region on 17, March 2010 for (a) wind speed (b) wind direction (c) Potential temperature (d) Temperature

-Reviewer's comment:

Is this weather a typical around Hyderabad. Also, how about the other days (8 January 2011 and 25 April 2011)?

-Authors' reply:

To examine this further we have plotted the meteorological fields over Hyderabad for the other days i.e. 8 January 2011 and 25 April 2011. The vertical profiles of wind speed, wind direction, temperature and potential temperature can be found in fig.4 for 8 January 2011 and fig.5 for 25 April 2011. It could well be noticed that the from fig.4a of the manuscript (fig.3a of this document), fig.4a and fig.5a that, the lower values of wind speed from surface to upto around 5-6km and gradual increase in wind speed beyond that, are seen as a consistent feature on all the 3 days over Hyderabad. Thus, such a feature looks to be a permanent character of winds over Hyderabad. Also, the prevalence of westerly to north westerly winds beyond 6-7 km also appears to be a common feature on all the 3 days (fig.3b, fig.4b and fig.5b).

The temperature and potential temperature on 17th March (fig.3d and fig.3c) show a different behavior vis-a-vis that during 8 January 2011 (fig.4c and fig.4d) and 25 April 2011 (fig.5c and fig.5d). While the temperature during March shows a decreasing trend from surface to around 4 km and an isothermal layer from 4 to 5 km (fig.3), the isothermal

[Figure]

**Figure 4.** Vertical profiles of meteorological variables obtained from balloon measurements during 8th January 2011, (a) wind speed (b) wind direction (c) Temperature (d) Potential Temperature

[Figure]

**Figure 5.** Vertical profiles of meteorological variables obtained from balloon measurements during 25th April 2011, (a) wind speed (b) wind direction (c) Temperature (d) Potential Temperature

layer of temperature is seen from surface to first 2km during January (fig.4c) and April (fig.5c) flights. The potential temperature in January (fig.4d) and April flights (fig.5d) also shows an increasing tendency throughout the vertical with a small layer around 2 km altitude with near-steady potential temperature, unlike the well mixed layer upto 4km during March (fig.3c). Thus, the winds fields show a consistent behavior during all the days, while temperature fields show some differences.

8. **Figs 5 and 6:**

   -Reviewer's comment:

   What is the difference in NoACEM/Ctrl between Fig 5 and Fig 6? I am confused.

   -Authors' reply:

   The figures 5 and 6 have now been combined with fig.1 and fig.2 of the manuscript. The modified figure can be found attached below (fig.6). In the modified figure NoACEM simulations means the model simulations without emissions of BC from aircrafts.

9. **Fig 7:**

   -Reviewer's comment:

   How much did the meteorological fields, i.e., air temperature, change by implementing the BC radiative impacts?

   -Authors' reply:

   The vertical profile of the changes in air temperature ($\Delta$T=T$_{ACEM}$-T$_{NoACEM}$) due to the prescription of BC emissions from aircrafts are plotted in fig.7. In addition to $\Delta$T, the vertical profiles of BC after the prescription of emissions from aircrafts are also plotted. The analysis is done for all the 3 balloon flight days. It could be well noticed that, the prescription of BC emissions from aircrafts increases the temperature in the vicinity of the elevated sharp BC layers (around 4 to 6 km and 10 to 12 km during March, around 10 to 12 km during January and around 4 to 6 km during April). Additionally, such a prescription also reduces temperature just below the sharp BC layer (just below 4 km during March, near the surface during January and near the surface and just above 4 km during April). Such increments and reduction in the temperatures could be due to the absorption of shortwave and long wave radiation by aircraft emitted BC. The magnitude of such increment and reduction in temperature appears to be around $\pm$ 0.2 to 0.3 K. However, it may be noted that, the model simulated BC magnitudes within the elevated sharp layers are on an average 5-6 times lower than that within the observed elevated sharp layers (fig.6), thus the atmospheric heating and the corresponding changes in temperature within the model simulations would also be proportionally lower vis-a-vis the reality.

**Technical Corrections**

1. **P4, L23:**

   -**Original sentence from the manuscript:**

[Figure]

**Figure 6.** Observed vertical profile of BC over Hyderabad obtained from balloon measurements (a). during 17 March 2010 (b). during 8 January 2011 (c). during 25 April 2011. The high altitude BC peaks in each profile are identified by letters P1 to P5. Observed vertical profile of dT/dz (K km-1 ) over Hyderabad region obtained from balloon measurements (d). during 17 March2010 (e). during 8 January 2011 (f). during 25 April 2011. The locations corresponding to high altitude BC peaks are identified by letters P1 to P5. Model simulated vertical profile of BC over the area in the vicinity of the balloon flight region for (blue line) NoACEM/Ctrl (control run) and (red line) ACEM (runs with prescription of aircraft BC emissions) during (g). 17 March 2010, (h). 8 January 2011 and (i).25 April 2011.

The WRF-Chem model used in this study employed horizontal grid spacing of 2 km with 70 vertical levels

-Reviewer's comment:

How do the authors define the 70-vertical layer? Please clarify each level.

-Authors' reply:

5  The model simulations have been carried out by specifying 70 vertical levels. The levels are specified as listed in the table 3 and are discussed below. Roughly, first 10 levels resolve the atmospheric boundary layer (i.e. upto 2km). Next 4 levels cover an altitude of 2km, with vertical spacing of 500m. Beyond 4km, next 16 levels are spaced with 100m vertical spacing and they reach upto a height of 6 km. Such a fine resolution has been set to resolve the sharp and confined observed layers BC within this altitude band. These layers are followed again by the layers with 500m vertical spacing within the

[Figure]

**Figure 7.** Change in temperature due to the prescription of BC emissions from aircraft within the model simulations. The left column shows the simulated vertical profile of BC, while the right column shows the changes in temperature due to the prescription of BC emissions from aircrafts. The upper panels show the scenario fro March 2010, the middle panel shows the scenario for January 2011 while the bottom panel is for April 2011.

altitude band of 6-8 km. These coarsely spaced layers are then followed by 2 zones (8-9 km and 10-11 km) of 1 km vertical depth with 100 m vertical resolution accompanied by 2 coarsely spaced layers (covering 9-10 km altitude band) in-between. Such a fine vertical resolution has been kept to resolve the sharp and confined layers of BC if simulated by the model. Beyond 11 km, 10 more layers are specified to cover the altitude upto 20 km.

2. **P8, L21:**
   **-Original sentence from the manuscript:**
   Additionally, we carried out one more model simulation, in which we prescribed the emissions of BC from biomass burning activities using FINN-version 1.5 inventory (Wiedinmyer et al., 2011) biomass burning data. This Fire INven-
10 tory from NCAR (FINN) provides high resolution, global emission estimates from such open burning activities.
   -Reviewer's Comment:
   The acronym "FINN" should be defined in L20 not L21.

**Table 3.** Vertical levels prescribed in WRF-Chem simulations

| Level Index | Altitude (km) |
|---|---|
| 01-10 | 0-2 |
| 11-14 | 2-4 |
| 15-31 | 4-6 |
| 32-37 | 6-8 |
| 38-49 | 8-9 |
| 50-51 | 9-10 |
| 52-59 | 10-11 |
| 60-70 | 11-20 |

- Authors' reply:

The manuscript will be modified with the corresponding correction.

3. **Fig 3:**

-Reviewer's Comment:

The map is too simple to know the information about city name and topography. Most of readers are not familiar to this area.

- Authors' reply:

Accepting the suggestions from both the reviewers, this figure will be removed from the manuscript.

**References**

[revised manuscript text omitted]

---

## Author Comment (AC2) · 22 May 2017

**Possible climatic implications of high altitude emissions of black carbon**

Gaurav Govardhan[1], Sreedharan Krishnakumari Satheesh[1,2], Ravi Nanjundiah[1,2], Krishnaswamy Krishna Moorthy[1], and Surendran Suresh Babu[3]

[1]Center for Atmospheric and Oceanic Sciences, Indian Institute of Science, Bengaluru, India
[2]Divecha Center for Climate Change, Indian Institute of Science, Bengaluru, India
[3]Space Physics Laboratory, Vikram Sarabhai Space Centre, Thiruvananthapuram, India

*Correspondence to:* Gaurav R. Govardhan (govardhan.gaurav@gmail.com)

**Authors' reply to comments from the second reviewer**

First of all, we would like to thank the anonymous reviewer for appreciating our work, giving constructive suggestions and overall positive recommendations.

**General comments from the reviewer:**

1. This study investigates the reasons behind measured high-altitude BC layers over India for 3 separate days using a chemistry transport model. The topic is relevant and within the scope of ACP. The introduction is clear and well-written. When it comes to the general presentation of the results and the figures, however, I think this manuscript needs further work before I can recommend it for publication. First, the aircraft emissions in this study were scaled, but this is only mentioned in one sentence and never mentioned again in the abstract or conclusion.

   -Authors' reply:

   We accept the aforementioned suggestion from the reviewer. The manuscript will be modified and the corresponding changes will be made.

2. The authors correctly states that the uncertainties in emissions are large, but it is not clear to me why the emissions were scaled, by how much, and how this could be justified.

   -Authors' reply:

20
   This point raised by the reviewer is similar to the point raised by the first reviewer in the third specific comment. The aircraft emissions of BC from MACCity inventory have been modified in this study by using published data regarding fuel efficiency of an aircraft, recent estimates of the emission factor of aircraft fuel for BC and the published data of air traffic over Hyderabad. The explanation to the modification/scaling of the emissions can be found in part 'b' of the

authors' reply to the third specific comment from the first reviewer.

3. Second, the manuscript is written like a story ('then we did this and then we did that'), and it needs to be rewritten to a more common form with 'Methods' and 'Results' clearly separated. I also miss a 'Discussion' section where the authors discuss some of the uncertainties with their study, e.g. the limitation of data (only 3 days), the scaling of the emissions etc. Third, the figures need to be plotted on the same format, and some of them can be combined or removed. Finally, I think there is a tendency for the interpretation of the results to be slightly overstated. The study itself is interesting, and it would read much better with a plain description of the results, without any 'convincing'. There are also use of words like 'significance difference' without any significance testing mentioned.

-Authors' reply:

We partly agree with the observation of the reviewer on the writing style and in the revised version we have changed it to a large extent taking cue from the comments. A discussion section has also been added in the revised version of the manuscript. To strengthen our results, we have carried out 1 month long model simulations (with prescription of aircraft BC emissions) for each of the 4 seasons. A brief discussion about the results of those simulations has been presented in the reply to the first general comment from the first reviewer. Those results will be discussed in greater details in the modified version of the manuscript. A few figures have also been combined into one figure for better understanding. Additionally, as suggested, wherever necessary, the statistical parameters like correlation coefficient, significance level have been computed.

**Specific comments from the reviewer:**

1. **Abstract**

   Reviewer's comment:

   Please specify the 3 dates for the measurements, and that you scaled the aircraft emissions.

   -Authors' reply:

   The corresponding modifications will be made in the abstract.

2. **Fig.1**

   Reviewer's comment:

   What does the error bars illustrate? Do you know why the error bars are much larger in a) and b) compared to c)?

   -Authors' reply:

   The error bars represent ensemble standard deviation. The balloon data is not equi-spaced along the vertical. The raw aethelometer data are averaged over regular altitude intervals combining the data obtained during the ascending and descending phases; and the bars represent the standard deviations representing the spread of the values in each ensemble.

[Figure]

**Figure 1.** Observed vertical profile of BC over Hyderabad obtained from balloon measurements (a). during 17 March 2010 (b). during 8 January 2011 (c). during 25 April 2011. The high altitude BC peaks in each profile are identified by letters P1 to P5.

Larger amplitudes of these bars represent larger fluctuation in the data. The bars are shorter in fig.1c, due to averaging over a larger vertical extent (a little reduced vertical resolution), because of the larger variation in the ascent speed of the balloon in that flight.

5 3. **Chapter 2:**

Reviewer's comment:

Can you add a small paragraph on how BC were measured using an Aethalometer, e.g. filter-based absorption. What are the uncertainties on separating out other absorbing components, like dust? Can you also report on the MAC that was used? I suggest specifying that 'elemental carbon' is measured (eBC), as suggested in literature (Petzold et al. (2013)

10 and references within). I suggest to drop Fig. 3 of the manuscript.

-Authors' reply:

We have replied to the reviewer's comment one by one.

-Reviewer's comment:

Can you add a small paragraph on how BC were measured using an Aethalometer, e.g. filter-based absorption. Can you

15 also report on the MAC that was used?

-Authors' reply:

The mass concentration of 'Equivalent Black Carbon' (EBC) was measured using an Aethelometer (Model AE-42, of

Magee Scientific, USA). The aethelometer estimates the mass concentration of EBC by measuring the change in the transmittance of a quartz filter to 880 nm light upon the deposition of aerosol. The value of effective Mass Absorption Cross-section (MAC) used in these measurements is 16 m$^2$ g$^{-1}$ (Hansen, 1996, 2005). The effective MAC includes the amplification of absorption due to multiple scattering on the filter fiber matrix and the decrease due to shadowing.

The aethalometer was configured for volumetric flow with an external pump, providing a flow rate of 14 liters per minute (LPM) at ground, and operated at a time base of 5 s. The data from the aethelometer was telemetered down along with the GPS co-ordinates. A few studies have reported uncertainties in the aethalometer estimated EBC (for e.g., Weingartner et al. (2003); Arnott et al. (2005); Sheridan et al. (2005); Corrigan et al. (2006)). Following the suggestions from the reports, we have used an amplification factor of 1.9 and an 'R' factor (shadowing effect) of 0.88 in this study. Nevertheless, when the aerosols are aged or mixed with others, as they would be away from the direct emissions, the shadowing effect would be negligible (Weingartner et al., 2003).

The modified manuscript will have the paragraph written above. The figure 3 has been removed from the manuscript.

-Reviewer's comment:

- What are the uncertainties on separating out other absorbing components, like dust?

-Authors' reply:

- The Mass Absorption Cross-section (MAC) for dust is two to three orders of magnitude lesser than the for BC (Hansen, 2005). Hence if the mass of dust is substantially higher than EBC, only then same optical absorption will be produced. Thus under normal conditions, the effects of dust on measured EBC mass concentrations would be negligible.

-Reviewer's comment:

- I suggest specifying that 'elemental carbon' is measured (eBC), as suggested in literature (Petzold et al 2013 and references within).

-Authors' reply:

- As suggested by the reviewer, we will follow the terminology specified by Petzold et al. (2013). The mass concentration measurements made using aethelometer will be reported as the 'Equivalent Black Carbon' (EBC) instead of 'Black Carbon' (BC).

4. Reviewer's comment:

There are many figures and I think it would be easier to read if some of the figures could be combined. For instance, may be Fig.1 and Fig.2 and Fig. 5/6 from the manuscript can be merged into one bigger figure (with 9 equally sized panels) so it is easier to compare the peaks? In that case; can you plot the panels on the same y axis, some goes up to 10 km vs. 9 km vs 12 km. I find it confusing that the axis and panels are plotted differently for each figure, e.g. comparing Fig.5 and Fig.6 from the manuscript, where the black line in Fig. 5 are the same as blue line in Fig.6 (?). I suggest dropping Fig.5 from the manuscript as the same info is in Fig.6. Fig 9 and Fig 10 from the manuscript can also be combined (with 9 panels).

-Authors' reply:

We agree with the modifications suggested by the reviewer. The corresponding suggestions are incorporated in the modified version of the manuscript. The modified figures can be found below as fig.2 and fig.3.

[Figure]

**Figure 2.** Observed vertical profile of BC over Hyderabad obtained from balloon measurements (a). during 17 March 2010 (b). during 8 January 2011 (c). during 25 April 2011. The high altitude BC peaks in each profile are identified by letters P1 to P5. Observed vertical profile of dT/dz (K km$^{-1}$) over Hyderabad region obtained from balloon measurements (d). during 17 March 2010 (e). during 8 January 2011 (f). during 25 April 2011. The locations corresponding to high altitude BC peaks are identified by letters P1 to P5. Model simulated vertical profile of BC over the area in the vicinity of the balloon flight region for (blue line) NoACEM/Ctrl (control run) and (red line) ACEM (runs with prescription of aircraft BC emissions) during (g). 17 March 2010, (h). 8 January 2011 and (i).25 April 2011.

5. **P6, L16:**

**Original sentence from the manuscript:**

While showing in general reduction in temperature with height similar to the observations (Red line, fig.4d), the model (Blue line, fig.4d) shows discrepancies in actual magnitudes vis-a-vis the observations, as in case of the other meteorological variables.

Reviewer's comment:

[Figure]

**Figure 3.** Vertical profiles of atmospheric heating rates for the observed BC profiles during (a).March 2010, (b). January 2011 and (c). April 2011. Vertical profiles of atmospheric heating rates corresponding to the observed BC profiles during (a).March 2010, (b). January 2011 and (c). April 2011. Vertical profiles of atmospheric heating rates corresponding to the model simulated BC profiles during (d and g). 17 March 2010 (e and h). January 2011 (f and i). 25 April 2011.

Can you report numbers for the differences in magnitudes you are referring to? It is hard to read on the figure.

-Authors' reply:

The model simulated vertical profile of temperature (Blue line, fig.4d) in-general shows similarity with the observed profile of temperature (Red line, fig.4d) retrieved from the thermometer attached as a pay-load to the balloon. However,
the two profiles differ in the exact magnitudes of the temperature. The differences in the observed and the simulated temperature magnitudes lie within -1.7 to +3.4 K.

6. **P6, L25:**

**Original sentence from the manuscript:**

The magnitudes of the simulated BC significantly match the observations only over the lower most altitudes i.e. below 3.5 km.

Reviewer's comment:

What do you mean by 'the simulated BC significantly match the observation'? Did you do any significance testing on

[Figure]

**Figure 4.** Comparison between the model simulated meteorological variables in NoACEM/Ctrl (control) run and the corresponding observations over the balloon flight region on 17, March 2010 for (a) wind speed (b) wind direction (c) Potential temperature (d) Temperature

this? Again, it is difficult to compare the model and the observations since the plots are so different (axis).

-Authors' reply:

During the comparison of the observed vertical profile of BC obtained from the balloon measurements with the model simulated vertical profile of BC in the control run (i.e. without the prescription of BC emissions from aircrafts), upon the visual inspection of the two profiles we notice that, the profiles roughly match each other within the lower atmosphere but differ drastically from each other beyond 3.5 km. The model does not depict the sharp elevated layers of BC (as seen in the observed BC profiles) in the simulations without the prescription of aircraft emissions of BC. So, in this comparison, we just wanted to highlight that, the rough agreements in the profiles are seen below 3.5 km and the disagreements in the profiles beyond 4 km altitude. Relatively better agreements between the model simulated vertical profile of BC and the observed BC profile are seen within 0.5 km to 2 km altitude band. Due to coarse resolution of model within this band (hence lesser number data points) vis-a-vis the observations, we could not find out the correlation coefficient between the two profiles and hence the significance of the correlation coefficient of this match could not be found out. Nevertheless, we have computed the ratio of the observed BC and the modeled BC all throughout the vertical for these profiles and same are shown in fig.5. It could be clearly seen that the ratio shows relatively lower values (ratio~3) upto 4 km altitude, specifically between the altitude band of 0.5 to 2 km (ratio ranges between 0.6 to 2). On the other hand, the ratio shows values greater than 10 beyond 4 km. Thus, we say that the model simulated BC shows better comparison with the observations below 3.5 km altitude. We will accordingly modify the sentence in the manuscript. The modified

figure (Fig.2) also has the modeled and observed vertical profiles of BC on the same vertical axis.

[Figure]

**Figure 5.** Ratio of the observed BC to the model simulated BC (in the control run)

7. **P7, L5:**

   **Original sentence from the manuscript:**

   The meteorology (observed as well as simulated) being benign for all the cases, and possibility of any long-range transport of BC from other location being an unlikely cause for such high concentrations

   Reviewer's comment:

   What do you mean by the meteorology being benign? And that the long-range transport of BC is unlikely?

   -Authors' reply:

   In this study, we first noticed that, the model WRF-Chem, simulated the meteorological parameters with satisfactory agreements with the corresponding observations obtained from the balloon measurements. In spite of such satisfactory agreements between the modeled and the observed meteorological parameters, the model fails to capture the high altitude sharp peaks in BC as observed. Thus we noticed that the local meteorology may not have played a major role in causing the observed sharp and confined high altitude BC peaks. Hence, we say that the local meteorology may not explain or be responsible for the observed sharp high altitude BC peaks.

   To assess the role played by long range transport of BC, Babu et al. (2011) used the Lagrangian back trajectory model, Hybrid Single Particle Lagrangian Integrated Trajectory HYSPLIT. The authors plotted the isentropic 7 day back trajectories of air parcel arriving over the balloon flight location at different vertical levels. It was found that, the air parcels beyond 5 km have origin in the African region or beyond that. It was also found that, BC showed very low concentrations in those altitudes up to around 8 km; before increasing again. Interestingly, between 8-9 km, BC showed one more

peak in concentration (Fig.2a), but the trajectories did not show any conspicuous shift. Thus, possibility of long-range transport in causing the peaks in BC was ruled out.

Thus, we rule out the local meteorology and the long-range transport as possible causes for the existence of such high altitude BC peaks. A citation will be provided to Babu et al. (2011) in this regards.

8. Reviewer's comment:

Your manuscript is written more like a story, and how you came about with the different hypothesis, but I suggest rewriting the manuscript with the more common separation of 'Methods' and 'Results', which means e.g. that 4.3 and 4.6 (until L15 on P10) should be moved to 'Methods' along with everything else in Results that explains the methods you have used. Parts of 4.7 explaining background for convective lifting can be moved to introduction (and the rest to methods). Same goes for Conclusions.

-Authors' reply:

We appreciate the aforementioned modifications suggested the reviewer and the corresponding changes in the layout of the manuscript will be made.

9. **P7 L32:**

**Original sentence in the manuscript:**

Thus, considering such underestimations of EI(BC) in the current emissions inventory, we modified the emissions and forced our model with such emissions at 23 levels with appropriate mapping to model levels.

Reviewer's comment:

This is the first time and only place you say that you have scaled the aircraft emissions you have used. How did you modify the emissions? You also need to mention this in the abstract/introduction, throughout the text and in the conclusions.

-Authors' reply:

This point raised by the reviewer is similar to the point raised by the first reviewer in the third specific comment. The aircraft emissions of BC from MACCity inventory have been modified in this study by using published data regarding fuel efficiency of an aircraft, recent estimates of the emission factor of aircraft fuel for BC and the published data of air traffic over Hyderabad. The explanation to the modification/scaling of the emissions can be found in part 'b' of the authors' reply to the third specific comment from the first reviewer.

As suggested by the reviewer, the modifications in the emissions will be mentioned in the manuscript.

10. **P9 L29:**

**Original sentence from the manuscript:**

This aerosol mixture has water soluble species (28000 # $cm^{-3}$ ), insoluble species (1.5 # $cm^{-3}$ ) and soot (130000 # $cm^{-3}$).

How is 'soot' defined in this context?

-Authors' reply:

The soot component considered in the Optical Properties of Aerosols and Clouds (OPAC) model is used to represent absorbing Black Carbon. The BC particles are not soluble in water and hence are assumed not to grow with relative humidity. The density of soot is taken as $1$ g cm$^{-3}$. The optical properties are calculated assuming the size distribution with many very small particles (which would have the density $2.3$ g cm$^{-3}$ ). More details about the soot component defined in OPAC can be found in Hess et al. (1998).

11. **P8, L7**

   -**Original sentence from the manuscript:**

   the nature of the layers (sharp and confined) looks very similar to those seen in the measurements

   -Reviewer's comment:

   'the nature of the layers (sharp and confined) looks very similar'. Please rewrite. Also, the peaks are much smaller in magnitude and are not located at the same altitudes as the observations?

   -Authors' reply:

   We acknowledge that the model simulated high altitude BC peaks are smaller in magnitudes vis-a-vis the observed high altitude BC peaks. Additionally, the model simulated peaks occur at different altitude as compared to the observation. Nevertheless, the overall pattern or character of the model simulated high altitude sharp and confined BC peaks resembles the observed high altitude BC peaks. Moreover, the correlation coefficients between the model simulated peaks and the observed peaks within the altitude band of the respective peaks comes out to be 0.8 and 0.97 for March and January profiles. The correlation coefficients are 95% and 99.9% significant respectively.

   -Modified sentence:

   the sharpness of the modeled BC layers make them look similar to the observed BC layers.

12. **P8, L11:**

   **Original sentence from the manuscript:**

   This clearly highlights the role played by aircraft emissions of BC in creation of the high altitude BC peaks

   -Reviewer's comment:

   'This clearly highlights the role played by aircraft emissions' However, you did scale the emissions? You need to emphasis this more. Do you see the peaks when you run with unscaled emissions?

   -Authors' reply:

   In this study, the aircraft emissions of BC from MACCity inventory have been modified by using published data regarding fuel efficiency of an aircraft, recent estimates of the emission factor of aircraft fuel for BC and the published data of air traffic over Hyderabad. The explanation to the modification/scaling of the emissions can be found in part 'b' of the authors' reply to the third specific comment from the first reviewer. The model does not capture the high altitude BC

peaks with unscaled emissions.

13. **P8, L34:**

    **Original sentence from the manuscript:**

    Beyond 4 km, the profiles are identical, implying that the elevated BC layers are insensitive to surface BC emissions.

    -Reviewer's comment:

    'Beyond 4 km, the profiles are identical'. It is hard to say this without any significance testing.

    -Authors' reply:

    As suggested by the reviewer, we have further analysed the model simulated vertical profiles of BC in the two cases i.e. with and without the surface emissions of BC. Specifically, we have compared the the two profiles beyond 4 km altitude. The correlation coefficient between the two BC profiles beyond 4 km comes out to be 0.97 which is 99.99% significant. Moreover, the the magnitudes of BC in the two profiles show good agreement with a difference limited to only $0.1\mu$g m$^{-3}$. This analysis suggests that the two profiles significantly match each other.

    Modified sentence:

    Beyond 4 km, the profiles are similar

14. **P9, L23:**

    **Original sentence from the manuscript:**

    During the winter flight (January 2011, fig.3b), the observed BC profile causes more heating near the surface than that at higher heights.

    -Reviewer's comment:

    : '..the observed BC profiles causes more heating near the surface'. Did you show that the observed BC profiles cause this heating?

    -Authors' reply:

    We in this study, compute the atmospheric heating rates caused by aerosol in general. In computation of the heating rates we use vertical profiles of the extinction coefficient obtained from CALIOP LIDAR on-board the CALIPSO satellite, over the Hyderabad region and the observed vertical profile of BC, using the methodology discussed in section 4.6.1 of the manuscript. Hence the heating rate profiles are not caused by BC alone but by the entire aerosol mixture. The corresponding changes in the manuscript will be made in this regards.

    Modified sentence:

    the observed BC profiles along with extinction coefficient profiles from CALIPSO, cause more heating near the surface

15. **P10, L29:**

    **Original sentence from the manuscript:**

    The identical nature of the heating rates profiles during March 2010 and April 2011, brings out the average features of heating rate profile during summer months over the region of study.

: 'The identical nature of (..), brings out the average pattern'. I do not understand this sentence.

-Authors' reply:

The vertical profiles of atmospheric heating rates due to observed BC profiles and the extinction coefficient profiles from CALIPSO during March 2010 (Fig.3a) and April 2011 (Fig.3c) show largely similar patterns. The correlation coefficient between the 2 profiles comes out to be 0.96 which is 99.99% . Such large similarities in the 2 profiles of heating rates over Hyderabad region, computed for 2 different years, highlight the consistent broad-features of the heating rate profiles over Hyderabad region during the pre-monsoonal period. Thus, we say that the identical pattern of the heating rate profiles during pre-monsoonal month (Fig.3a and Fig.3c), bring out the broad features of the the pre-monsoonal heating rate profile.

Modified sentence:

The largely similar nature of the heating rates profiles during March 2010 and April 2011, brings out the average features of heating rate profile during summer months over the region of study.

16. **P11, L21:**

**Original sentence from the manuscript:**

We entitle 'starting point of the trail' as that area in the aircraft emitted BC trail, where emission magnitudes are substantially higher than that over the remaining part of the trail

Reviewer's comment:

I am not sure if I understand your explanation of 'normal' and 'extreme' profiles in terms of the starting point. Area in the model?

Authors' reply:

The high altitude emissions of BC from aircrafts are in the form of a narrow trail. The area where two or more such trails cross each other (mainly in the vicinity of an airport) show higher magnitudes of emissions than the remaining part of the trail. So, we mainly refer to these areas when we use the phrase 'starting point of the trail'. Nevertheless, to avoid such confusion, the use of the phrase 'starting point of the trail' has been avoided. Instead, we simply specify such area as the one which shows higher magnitudes of aircraft emissions relative to the remaining part of the trail.

17. **Fig. 13 from the manuscript:**

Reviewer's comment:

Fig. 6 is very difficult to read.

-Authors' reply:

We have made modifications in the caption of the figure, for better understanding. The modified caption can be found with the figure 6 and is written in this reply as well.

[Figure]

**Figure 6.** The time-series of the vertical profile of extinction coefficient at stratospheric altitude, stratospheric AOD and the vertical profile of particle depolarization ratio at stratospheric altitude are plotted for each of the 5 regional boxes shown in fig.12 (previous figure from the manuscript). The time-series are plotted from January 2010 to December 2012. The time-series of the aforementioned parameters for each region are joined to each other to form a single time-series for each parameter. Such appended time-series is shown for a) vertical profile of extinction coefficient at stratospheric altitude b) stratospheric AOD and c) the vertical profile of particle depolarization ratio at stratospheric altitude . The letter 'M' signifies the monsoon season (June-July-August-September) embedded within the time-series for each region. In 'b', the background stratospheric AOD for the entire tropical belt (Kremser et al., 2016) is shown by the dotted red line.

Modified caption:

The time-series of the vertical profile of extinction coefficient at stratospheric altitude, stratospheric AOD and the vertical profile of particle depolarization ratio at stratospheric altitude are plotted for each of the 5 regional boxes shown in fig.12 (previous figure from the manuscript). The time-series are plotted from January 2010 to December 2012. The time-

5          series of the aforementioned parameters for each region are joined to each other to form a single time-series for each parameter. Such appended time-series is shown for a) vertical profile of extinction coefficient at stratospheric altitude b) stratospheric AOD and c) the vertical profile of particle depolarization ratio at stratospheric altitude . The letter 'M' signifies the monsoon season (June-July-August-September) embedded within the time-series for each region. In 'b', the background stratospheric AOD for the entire tropical belt (Kremser et al., 2016) is shown by the dotted red line.

10     **Technical corrections:**

1. **P2, L1**

   **Original sentence from the manuscript:**

   Several earlier studies have shown that such an atmospheric heating by a layer of aerosol with high BC abundance

   -Reviewer's comment:

   What is 'such an' refer to here? I suggest to remove.

   -Authors' reply:

   We follow the suggestion from the reviewer. The words 'such an' have been removed from the text.

2. **P2, L12:**

   **Original sentence from the manuscript:**

   Additionally, the atmospheric heating at higher altitude can give rise to local stable scenario below, which can affect convection and consequently impact precipitation.

   -Reviewer's comment:

   Can you provide references to precip/convection?

   -Authors' reply:

   The surface cooling and atmospheric warming due to black carbon can alter the vertical profiles of temperature, induce more stability and reduce convection and the resulting rainfall. Fan et al. (2008) discusses these effects of BC on convection and rainfall in greater details. The study will be cited in the modified manuscript.

3. **P6, L3:**

   **Original sentence from the manuscript**:

   up to $1^{st}$ 2 km of the lower troposphere

   -Reviewer's comment:

   $1^{st}$ -first

   -Authors' reply:

   The corresponding changes will be made in the manuscript.

4. **P6, L8:**

   **Original sentence from the manuscript**:

[revised manuscript text omitted]

---

## Author Response (AR2)

**Possible climatic implications of high altitude emissions of black carbon**

Gaurav Govardhan[1], Sreedharan Krishnakumari Satheesh[1,2], Ravi Nanjundiah[1,2], Krishnaswamy Krishna Moorthy[1], and Surendran Suresh Babu[3]

[1]Center for Atmospheric and Oceanic Sciences, Indian Institute of Science, Bengaluru, India
[2]Divecha Center for Climate Change, Indian Institute of Science, Bengaluru, India
[3]Space Physics Laboratory, Vikram Sarabhai Space Centre, Thiruvananthapuram, India

*Correspondence to:* Gaurav R. Govardhan (govardhan.gaurav@gmail.com)

**Authors' reply to comments from the first reviewer**

At the outset, we thank the co-editor and the anonymous reviewers for meticulous and scholarly evaluation of the manuscript and giving useful suggestions.

**Comments from the reviewer:**

I think this revised manuscript will be accepted, after two points, i.e., treatment of BC and vertical levels in the model, are also explained in the manuscript. These points are explained in the Author's Response, but they are not inserted into the revised manuscript. So I just ask the authors to add them to the updated manuscript.

10 Authors' reply:

We have made the following necessary modifications in the manuscript.

- Reviewer's comment:

1) As for the treatment of BC in the model (their answer of 2 in the specific comments from the reviewer), these two sentences are important; "The conversion of e-folding lifetime from hydrophobic to hydrophilic is considered to be 2.5 days. More details

15 about the treatment of BC in WRF-Chem can be found in Kumar et al. (2015)".

-Authors' reply:

This description about treatment of BC in the model has been added into the modified version of the manuscript. The modification could be found at P5 L14-15 in the track changed and the modified version of the manuscript.

20 - Reviewer's comment:

2) As for the vertical levels in the model (Table 3 in the Author's Response), the information must be also inserted into the uploaded manuscript.

-Authors' reply:

The Table 3 in the authors' response has been added into the main manuscript as Table 1, with the relevant description at P4

25 L31 to P5 L1 in the track changed and the modified version of the manuscript.

**References**

Kumar, R., Barth, M. C., Pfister, G. G., Nair, V. S., Ghude, S. D., and Ojha, N.: What controls the seasonal cycle of black carbon aerosols in India?, Journal of Geophysical Research: Atmospheres, 120, 7788–7812, doi:10.1002/2015JD023298, http://dx.doi.org/10.1002/2015JD023298, 2015JD023298, 2015.

**Possible climatic implications of high altitude emissions of black carbon**

Gaurav Govardhan[1], Sreedharan Krishnakumari Satheesh[1,2], Ravi Nanjundiah[1,2], Krishnaswamy Krishna Moorthy[1], and Surendran Suresh Babu[3]

[1]Center for Atmospheric and Oceanic Sciences, Indian Institute of Science, Bengaluru, India
[2]Divecha Center for Climate Change, Indian Institute of Science, Bengaluru, India
[3]Space Physics Laboratory, Vikram Sarabhai Space Centre, Thiruvananthapuram, India

*Correspondence to:* Gaurav R. Govardhan (govardhan.gaurav@gmail.com)

**Authors' reply to comments from the second reviewer**

At the outset, we thank the co-editor and the anonymous reviewers for meticulous and scholarly evaluation of the manuscript and giving useful suggestions.

**Comments from the reviewer:**

I appreciate the authors' effort to make the suggested changes to the manuscript. It reads much better and the methods are well explained.

I recommend publication if the authors could change 'realistic emissions' in the abstract and the conclusions (and 3 other
10   places) with 'a new emission inventory' or similar.

-Authors' reply:

We have made the corresponding changes.

**In Abstract:**

15   Original statement:

Our study demonstrates that, the high-flying aircrafts (with realistic emissions) are the most likely cause of these elevated BC layers.

Modified statement:

Our study demonstrates that, the high-flying aircrafts (with emissions from the regionally fine-tuned MACCity inventory) are
20   the most likely cause for these elevated BC layers.

The corresponding changes could be found at P1 L7-8 in the track changed and the modified version of the manuscript.

**In Conclusion:**

Original statement:

Upon the prescription of realistic BC emissions from aircrafts the model simulated vertical profile of BC started showing the
25   mysterious high altitude BC peaks as seen in the observations.

Modified statement:

Upon the prescription of aircraft BC emissions from the regionally fine-tuned MACCity inventory the model simulated vertical profile of BC started showing the mysterious high altitude BC peaks as seen in the observations.

The corresponding changes could be found at P19 L19-20 in the track changed and the modified version of the manuscript.

**At 3 other places:**

1. Original statement:

We noticed that the model, WRF-Chem, produces the elevated sharp peaks of BC akin to the observations upon the prescription of realistic emissions of BC from aircrafts.

Modified statement:

We noticed that the model, WRF-Chem, produces the elevated sharp peaks of BC akin to the observations upon the prescription of aircraft BC emissions from the regionally fine-tuned MACCity inventory.

The corresponding changes could be found at P12 L10 in the track changed and the modified version of the manuscript.

2. Original statement:

Thus, with the help of a regional chemistry transport model, our study showed that the aircrafts (with realistic emissions) appear to be one of the primary causes behind the occurrence of the the sharp elevated layers of BC over the Indian region.

Modified statement:

Thus, with the help of a regional chemistry transport model, our study showed that the aircrafts (with emissions from the regionally fine-tuned MACCity inventory) appear to be one of the primary causes behind the occurrence of the sharp elevated layers of BC over the Indian region.

The corresponding changes could be found at P18 L22 in the track changed version of the manuscript and at P18 L21-22 in the modified version of the manuscript.

3. Original statement:

The BC emissions from aircraft, in the MACCity inventory were scaled in this study, to match realistic emissions of BC from aircrafts over the Indian region.

Modified statement:

The BC emissions from aircraft, in the MACCity inventory were scaled in this study, with necessary modifications.

The corresponding changes could be found at P18 L30-31 in the track changed and the modified version of the manuscript.

[revised manuscript text omitted]